# Sulfide and Fluoride Mineralization of the NNE Region of Achemmach (Central Morocco): Paragenetic Sequences and Pyrrhotite-Sphalerite Geothermometry Constraints

**Hafid Mezougane** [1,2,*], **Mohamed Aissa** [2], **Souiri Muhammad** [2,3], **Azizi Moussaid** [2], **Abdelaziz El Basbas** [4], **Mourad Essalhi** [5], **Abdel-ali Kharis** [6], **Mohammed El Azmi** [2,3], **Ahmed Touil** [7] **and Essaid Bilal** [8]

1  Physico-Chemistry of Processes and Materials Laboratory, Research Team Geology of the Mining and Energetics Resources, Faculty of Sciences and Techniques, Hassan First University, Settat 26002, Morocco
2  Department of Geology, Faculty of Sciences, Moulay Ismail University, M.B. 11201, Zitoune, Meknes 50070, Morocco; mohamedaissa@gmail.com (M.A.); m.souiri@edu.umi.ac.ma (S.M.); moussaid.azizi@gmail.com (A.M.); m.elazmi@managemgroup.com (M.E.A.)
3  Managem Group, Twin Center, Tour A, Angle Boulevards Zerktouni et Al Massira Al Khadra, M.B. 5199, Casablanca 20250, Morocco
4  National Higher School of Mines of Rabat (ENSMR), M.B. 753, Agdal, Rabat 10000, Morocco; abdeazizelbasbas@gmail.com
5  Faculty of Sciences and Techniques, Moulay Ismail University, M.B. 509, Boutalamine, Errachidia 52003, Morocco; m.essalhi@umi.ac.ma
6  Polydisciplinary Faculty of Ouarzazate, Ibn Zohr University, M.B. 45000, Agadir 80000, Morocco; a.kharis@uiz.ac.ma
7  Department of Geology, Faculty of Sciences and Techniques, Cadi Ayyad University, M.B. 549, Av. Abdelkarim Elkhattabi, Gueliz, Marrakech 40000, Morocco; a.touil@uca.ma
8  Campus of Saint-Etienne 158, National School of Mines of Saint-Etienne (ENSM-SE), Cours Fauriel, 42000 Saint-Etienne, France; ebilal@emse.fr
*  Correspondence: hafid.mezougane@uhp.ac.ma

**Abstract:** Sulfide and fluoride mineralization in the NNE Achemmach (NNE-A) area is located in the NE of Central Hercynian Morocco. In veins or when disseminated, it is hosted either in Visean sedimentary formations or in the magmatic bodies, described for the first time in this article and corresponding to pillow-lavas, dolerites and olivine-bearing gabbros. The mineralization is multiphase and results from the succession of the following three events: (i) an early high-temperature hydrothermal event (T ≈ 350–420 °C) associated with a simple primary sulfide paragenesis composed of pyrrhotite, pyrite, chalcopyrite, sphalerite and galena with gangue of quartz.(ii) The second event corresponds witha low temperature fluorite hydrothermal one (T≈ 120–160 °C), whereas the (iii) third is marked by, the deposition of a late sulfide paragenesis in a carbonate gangue within a moderate temperature (T≈ 200–250 °C). The temperatures of the paragenetic stages (350–400 °C) are estimated on the basis of the geothermometry constraints of the mineralogical assemblages, particularly the pyrrhotite-sphalerite equilibrium, in which the FeO content varies from 9.23 to 14.42 Wt%, and in the full study of their corresponding fluid phases. They are in perfect agreement with the fluid inclusion data of the first event.

**Keywords:** sulfideand fluoride mineralization; pyrrhotite-sphalerite; geothermometry; fluid phases; NNE of Achemmach; central Morocco

## 1. Introduction

The NNE sector of Achemmach (NNE-A) is situated 40 km southwest of Meknes city and about 5 km to the east of the El Hammam fluorite mine. It is in the northeastern part of the Morocco Central Hercynian (MCH), which constitutes one of the main units of the Mesetian domain corresponding to the main orogenic zone of the Moroccan Variscan belt (Figures 1 and 2). In addition, this domain is characterized by the occurrence of a

set of magmatic intrusions in a very contrasting metamorphic environment. The main granitic bodies outcroppings in this area are represented by Zaër, Ment and OulmèsVariscan granites. Other small outcrops also exist in this area such as the Moulay Bouazza, Aouam and El Hammam bodies. Generally, these magmatic plutons (granites), dikes and sills (microgranites and dolerites), or volcanic rocks (basaltic pillow-lavas) are intercalated within the sedimentary formations. This magmatic activity has generated the deposition of numerous types of economic mineralization.

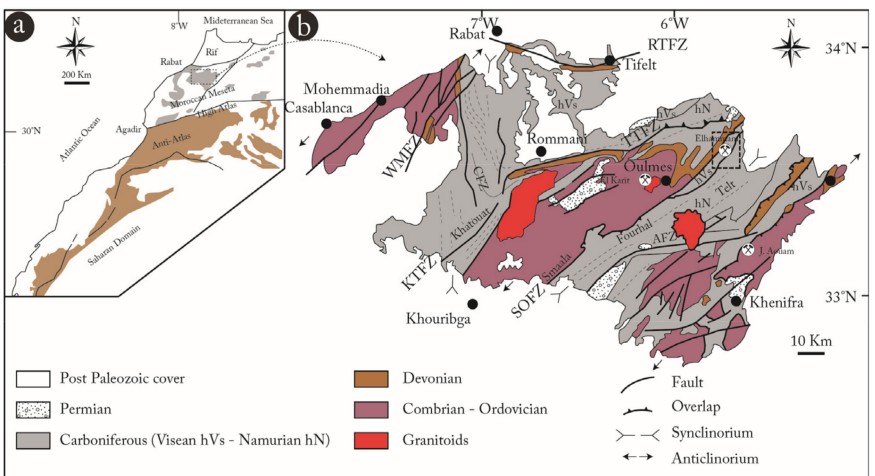

**Figure 1.** Structural domains of Morocco and location of the Central Morocco domain (**a**), and its main structural units [1,2] (**b**). (WMSZ: West Mestian Shear Zone, RTFZ: Rabat-Tiflet Fault Zone, SOFZ: Smaala-Oulmes Fault Zone, TTFZ: Tsili-Tfoudeit Fault Zone, AFZ: Aguelmous Fault Zone, KTFZ: Kef Tallal Fault Zone, CFZ: Cherrat Fault Zone).

This economic mineralization contains Sn, W, Au, Pb, Zn, Ag, Sb, F, and Ba chemical elements, where JbelAouam Pb-Zn-Ag deposit, El Hammam Fluorite deposit and Achemmach Sn ore deposit arethe main exploited ore deposits. The principal features of these mineralizations are as follows: (1) Tin and tungsten mineralization are hosted by the Hercynian granitoids, with two types of mineralizations, such as the scheelite-bearing skarn type discovered at the JbelAouam and El Hammam deposits, and mineralization related to an acidic-to-hydrothermal stage. Thelatter is identified at (i) the El Karit deposit with an assemblage of cassiterite, scheelite, wolframite, and molybdenite-bearing quartz veins. Closed in 1973,ElKarit was the largest mine in central Morocco, with production estimated at 650 t of cassiterite; (ii) at the Achemmach deposit, where cassiterite is hosted by tourmaline filling structures; (iii) at theZguit to Oulmès areas, where mineralization is marked by several stanno-wolframiferous occurrences, and (iv) around the Ment and Zaër granites, where Sn-W pyrometasomatic concentrations are found. (2) Zaër are dispatched in different structural offshoots, including the following: polymetallic mineralization consisting of Zn, Pb and Ag vein deposits. This mineralization is associated with quartz, siderite, ankerite, andless commonly, barite gangue minerals. This polymetallic mineralization is structurally controlled by the main regional faults at JbelAouam, Moulay BouAzza, JbelKhetem, and Ain Koheul. The JbelAouam Pb (Ag) ore deposit, currently in production attheTouissit Mining Company (known in French as "Compagnie Minière de Touissit" or CMT) produces around 25,000 tof Pb and 1400 t of Agannually. (3) Antimoniferous mineralization occurs in numerous veins or as impregnations within the Visean formations of Mrirt and Kef N'sour [3,4]. Both Timekhdoudine (Kef N'Sour) and Tourtit-MasserAmane (Mrirt region) representthe most important sites of antimony with an estimated production of around 25,000 t. (4) Barium mineralization occurs in the following two forms: veins associated with fluorite, Pb-Zn sulfides and antimony, or as impregnations in Visean carbonates (Guertila, Bou Oussel, and Bertki). The barite minerals were exploited by many ancient artisanal mining operations. (5) Iron mineralization is performed at the Ait Amar, Tiflet, Boulhaiut

and Jbel Lahdid ore deposits. The main iron deposit was the Ait Amar, which is hosted by the oolitic ironstones of the Ordovician formations. An estimated 4,000,000 t was extracted between 1937 and 1955 and (6) Fluorite mineralization is recognized in some veins linked to regional shear zones. The most important fluorite occurrences were observed at the El Hammam vein deposit and at the Jbel Zrahina vein deposit in which fluorite was associated with barite.

The El Hammam fluorite mine operated by the Managem mining group was the first economic fluorite production site in the Moroccan Central Massif. The study area, situated at the NNE of the Achemmach deposit, contains several mineral parageneses, mainly formed bypyrite, pyrrhotite, marcasite, chalcopyrite, galena, sphalerite, ullmannite, hematite, magnetite and monazite. Relationships between these mineralizations, quartzitic sandstone and magmatic rocks (dolerites, gabbros, and pillow-lavas) clearly exist. A number of metric veins and centimetric vein letsare developed and filled by these minerals; especially pyrite, pyrrhotite, sphalerite, galena and chalcopyrite. This type of sulfide mineralization, witha laminated texture, is hosted in green schists as shown by core-drill interpretations [5]. Undoubtedly, fluorite veins are also present within this newly explored area (NNE-A).

The main goal of this work is to elucidate the spatio-temporal and genetic relationships that could exist between the sulfide–fluoride mineralization and the magmatic host rocks, recognized for the first time in the NNE-A sector, and their possible relationship with the hypothetical Achemmach granite.

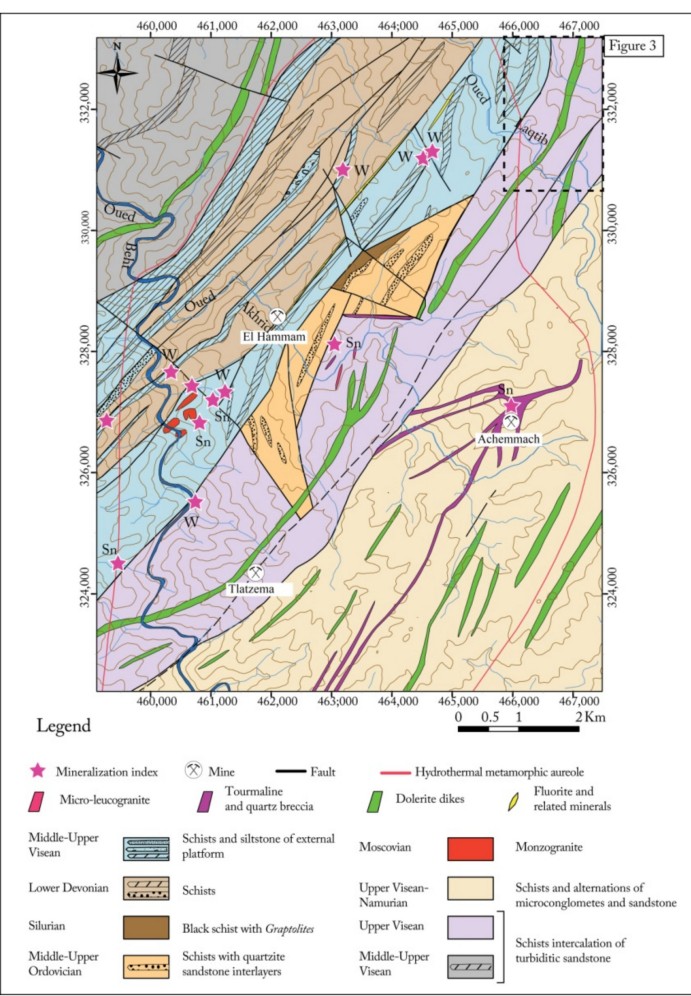

**Figure 2.** Geological map of the Achemmach-El Hammam district [6], Sn: Tin, W: Tungsten.

## 2. Geological Setting

The NNE-A belongs the Variscan Moroccan Central Massif. It consists of a local reduced Neoproterozoic bedrock, represented by acidic and intermediate volcanic rocks (rhyolites and andesites, respectively) and plutonic ones (granites) [7,8], a Palaeozoic basement (Cambrian to Permian) and Meso-Cenozoic cover. The Cambrian formations are composed of a carbonates-based platform and volcano-detrital sediments. The Ordovician is represented by detrital deposits (shales, sandstones and quartzites) in a shallow marine platform-type environment [9,10]. The Silurian is characterized by a glacio-eustatic transgression, producing black graptolitic pelites and carbonates. The Lower and Middle Devonian are dominated by clay and carbonate minerals from a reef deposit [2,11–24].

Devono–Dinantian Eovarisc deformation is responsible for the formation of both the Sidi Bettache and Azrou-Khenifra basins; first, as a pull-apart controlled by NNE-SSW to NE-SW border offsets, then, asaafore land type controlled by systems of folds and overlaps towards the WNW [6,16,19,21,25–28]. Carboniferous sedimentation is mainly represented by the Visean and Namurian deposits. In central Morocco, the Visean is made up of thick detrital and carbonate series [15,18–20,27,29–32], resulting in conglomerates, schists and limestones followed by sandstone and pelitic formations. The Namurian is mainly composed of flysches [29,31]. These Visean and Namurian deposits are associated with lava and basic intrusions (basalts, dolerites and gabbros) [15,26,33–37]. Thismagmatism shows a transitional alkaline to tholeiitic affinity [36] and locally a calc-alkaline one inthe NE of the Fourhal region [36].

The Lower Permian (Autunian) formationsareformedby conglomerates, sandstones, siltstones and cinerites in an intra-continental basin, and are associated with significant calc-alkaline volcanism (trachytes, andesites and rhyolites) [29,38–43].

The post-Paleozoic cover begins with Triassic conglomerates, sandstones, red argillites, dolerite and basalts [44–46]. These magmatic rocks are dated from $201.7 \pm 2$ to $197.8 \pm 0.7$ Ma [46]. Developed in the southern part of the massif, the Upper Cretaceous deposits (marls and limestones) [1] directlyoverlap with the Triassic rocks (lack of Jurassic, Lower and Middle Cretaceous formations). At the end of the Upper Cretaceous and the Eocene, phosphate layers were formed (Plateau des Phosphates).

The major Hercynian deformation (Upper Westphalian-Stephanian), dated between 300 Ma and 290 Ma [47], which responds to NW-SE compression, structured the Moroccan Central Massif into the NE-SW oriented anticlinorial and synclinorial units [1,16,19,22]. These ductile structures are separated by mega faults, including the Smaala–Oulmès Fault Zone (SOFZ), which extends from the Smaala in the south [38,48,49] to El Hammam in the north [29,31,34,50]. These structural units show syn-schist folds, generally oriented to the SE and locally to the NW, with associated overlaps and thrusts [22]. In this Mesetian domain, syn-tolate-tectonic granitic plutons [51–55] are calc-alkaline and mostly peraluminous [56–58].

## 3. Methodology

Geological mapping of the NNE-A was carried out in collaboration with the geological service of SAMINE mining company (Managem group), which allowed us to establish a geological map and a detailed lithological column and to define the host-rocks of the sulfide mineralization. In order to establish the mineralogical assemblage and paragenetic sequence of this ore deposit, 100 samples were collected from mineralized structures. A total of 77 polished sections were prepared and studied under the polarized microscope (transmitted and reflected lights) at the Department of Geology, Faculty of Sciences, Moulay Ismaïl University (Meknès, Morocco). This microscopic investigation was coupled with scanning electron microscopy (SEM) at the University of Quebec in Montréal (Canada) to analyze the following elements: Cu, Fe, S, Co, Ni, As, In, Ag, Pb, Sb and Cd. The analysis conditions were15 kV, 20 nA, and 20 s for all elements. The following standards and emission lines were used: pyrite (Fe K$\alpha$ and S K$\alpha$), metallic Co (Co K$\alpha$), metallic Ni (Ni K$\alpha$), chalcopyrite (Cu K$\alpha$), synthetic sphalerite (Zn K$\alpha$), metallic Ag (Ag L$\alpha$), synthetic

galena (Pb Mα), GaAs (As Lα), metallic Cd (Cd Lα), stibnite (Sb Lα). Microthermometric analyses of fluid inclusions were carried out on double-polished sections (150–200 μm) of gangue minerals (quartz, fluorite and dolomite) at the Department of Geology, Faculty of Sciences, Moulay Ismaïl University (Meknès, Morocco).

## 4. Results

### 4.1. Local Geology Field Work

The studied area is formed by the following two Paleozoic rock-types: (i) a meta-sedimentary sequence, composed of Middle Visean limestone and shale-sandstone and flyschoid of the Upper Visean-Namurian age, and (ii) a magmatic rock one represented by volcanic (pillow-lavas) and hypovolcanic rocks (dolerites), and olivine-bearing gabbros (Figures 3–5) [29].

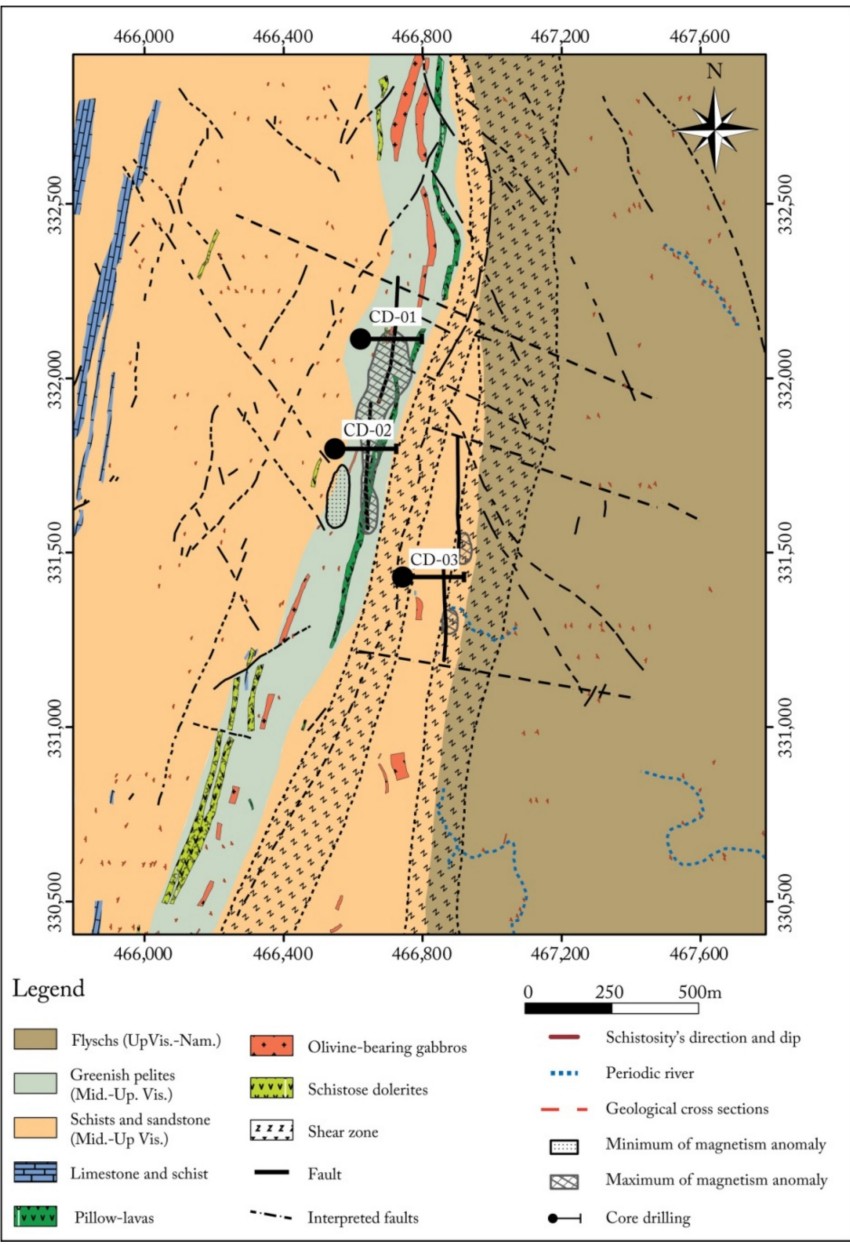

**Figure 3.** Geological map of the NNE-A (Mid.-UpVis.: Middle-Upper Visean, UpVis-Nam.: Upper Visean-Namurian).

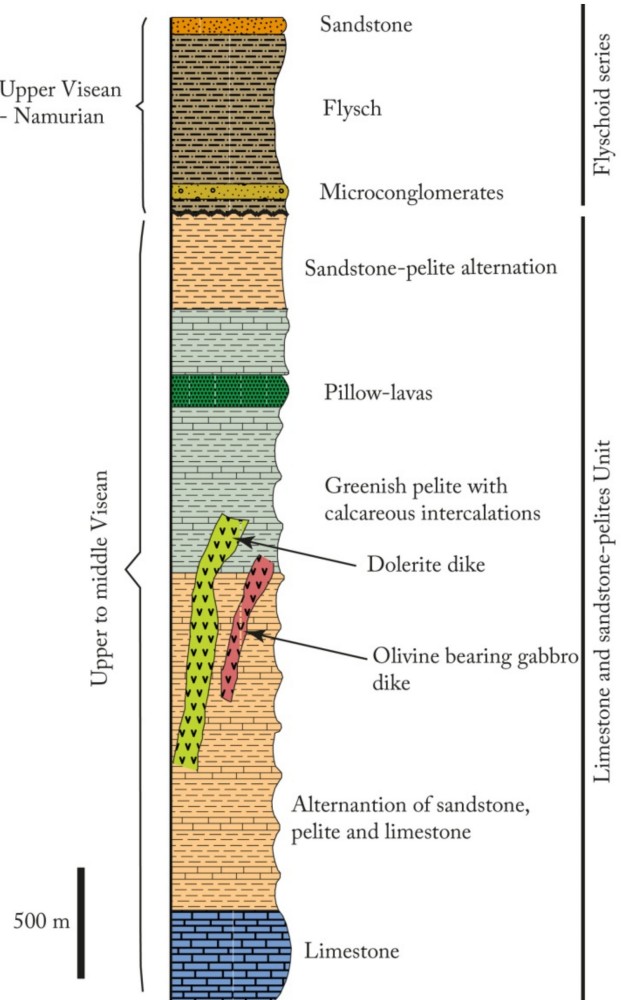

**Figure 4.** Synthetic lithostratigraphic column of the NNE-A.

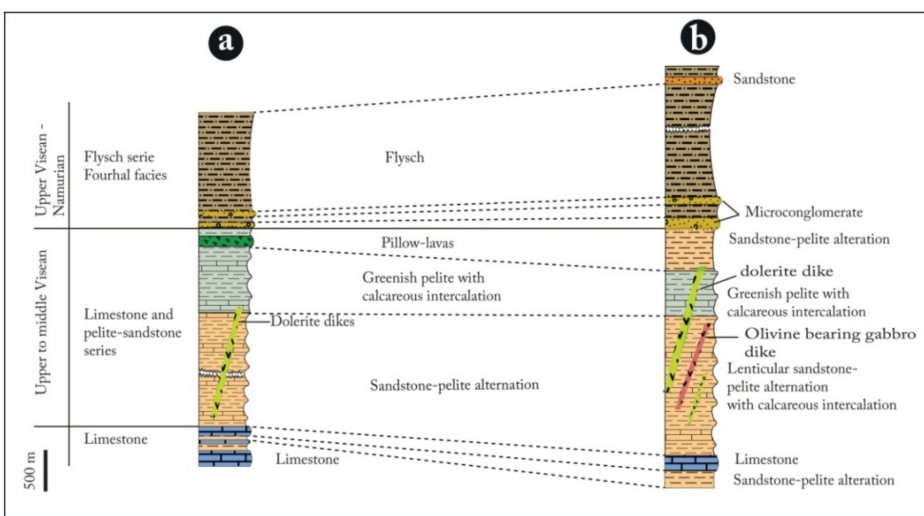

**Figure 5.** Correlation between synthetic lithostratigraphic logs of the southern (**a**) and northern (**b**) sectors of study area.

Based on the emplacement model, the structural framework, the relative chronology, and the petrogeochemistry of the magmatic rocks, we distinguished the following for the first time in this area:

(i)   Decimeter to meter-thick greenish pillow-lavas, with sharp borders and radius fractures underlined by fine greenish pelitic sedimentary intercalations, indicating recurrent volcanic activity in short episodes (Figure 6). Plagioclases and pyroxenes (augite) microlites, and rare phenocrystals are recognizable in a glassy matrix devoid of recognizable olivine (Figure 7);

(ii)  Deformed, metamorphosed and altered dolerite dikes crossingthe Middle to Upper Visean shale-sandstone formations (Figure 8) with an overall NE-SW direction with a NW dip-direction. They have a microlithicmesostasis texture and are composed of sericitized plagioclases in close association with amphibolitized pyroxenes, tourmaline with a variable degree of chlorite substitution, rutile, and opaque minerals (Figure 9);

(iii) Olivine-bearing gabbros, outcropped in variable dimensions (few meters to 20 m). These plutonic bodies have a granular texture and are mainly made of plagioclase, pyroxene, olivine, sphene, rutile, apatite and opaque minerals. All their constituting minerals were affected by different degrees of replacement by secondary minerals; the plagioclase was sericitized and albitized, while pyroxene was amphibolitized and epidotized, and olivine was serpentinized and chloritized (Figure 10) [29].

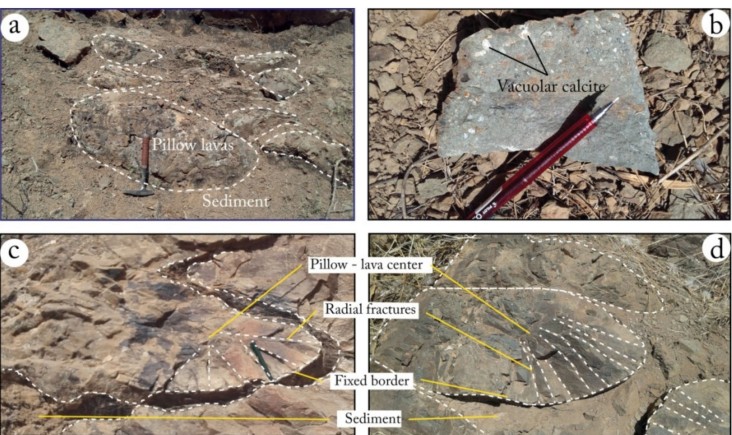

**Figure 6.** (**a**) Pillow-lavas, (**b**) vacuolar basalt, (**c**) detailed pillow structure showing radial fractures and (**d**) pillow-lava in contact with greenish pelites.

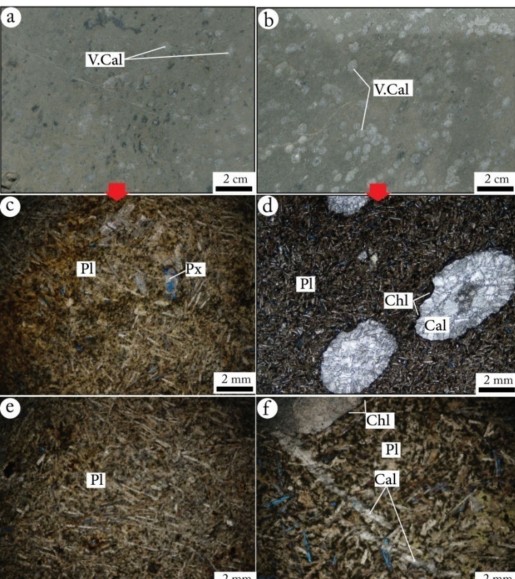

**Figure 7.** Macroscopic view of pillow-lavas less riche (**a**) andvery rich (**b**) in vacuoles of calcite, and microphotographs of this facies in transmitted light with crossed nicols (**c–f**). V.Cal: vacuoles of calcite, Pl: plagioclase, Cal: calcite, Chl: chlorite, Px:pyroxene.

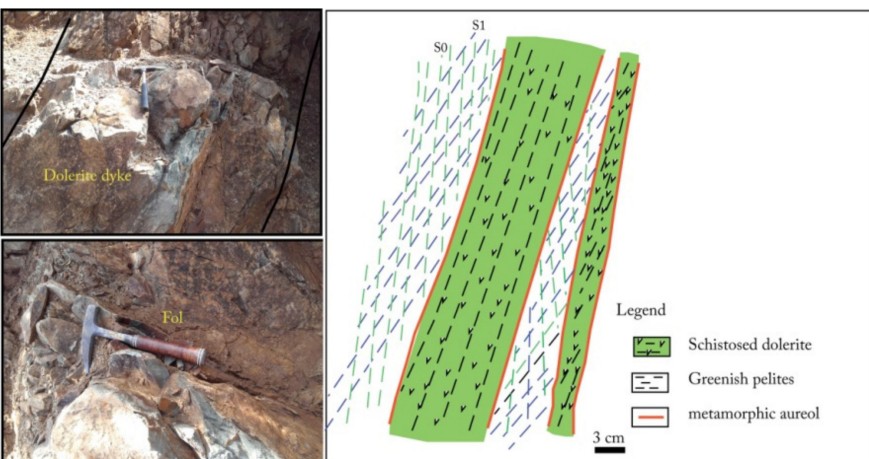

**Figure 8.** Foliation (Fol) in outcropping dolerites.

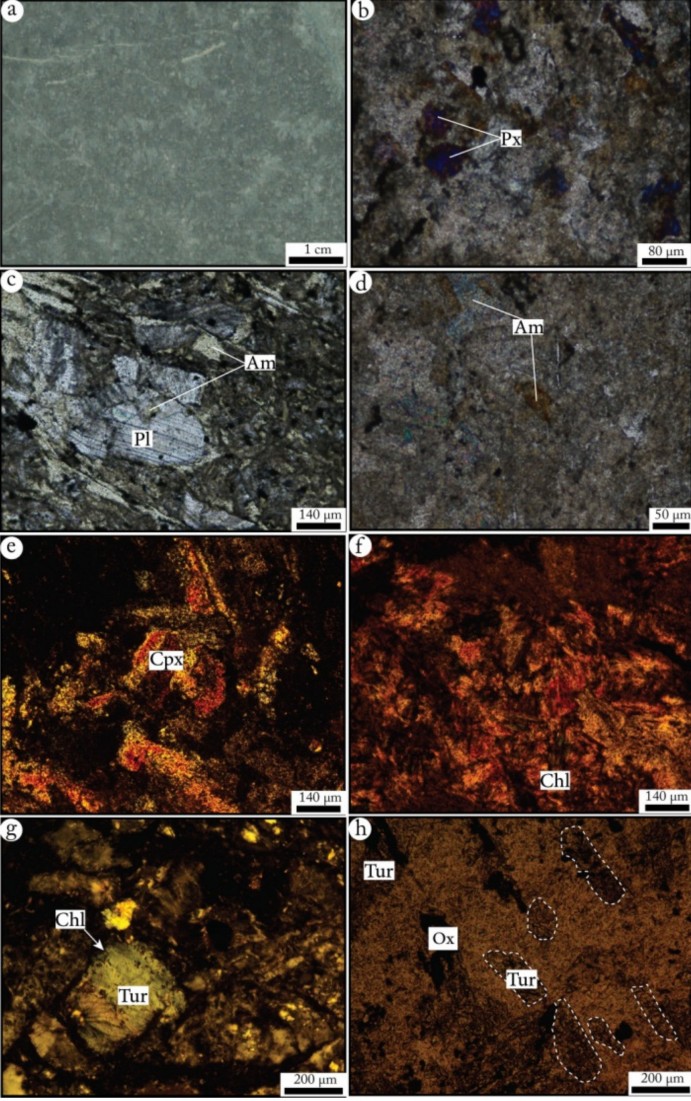

**Figure 9.** (**a**) Macroscopic view of a dolerite facies. (**b–h**): microphotographs of dolerites: plagioclases (Pl), amphibolitized (Am) pyroxenes (Px), clinopyroxenes (Cpx), chloritized (Chl) tourmaline (Tur) and oxides (Ox) (transmitted light—crossed nicols).

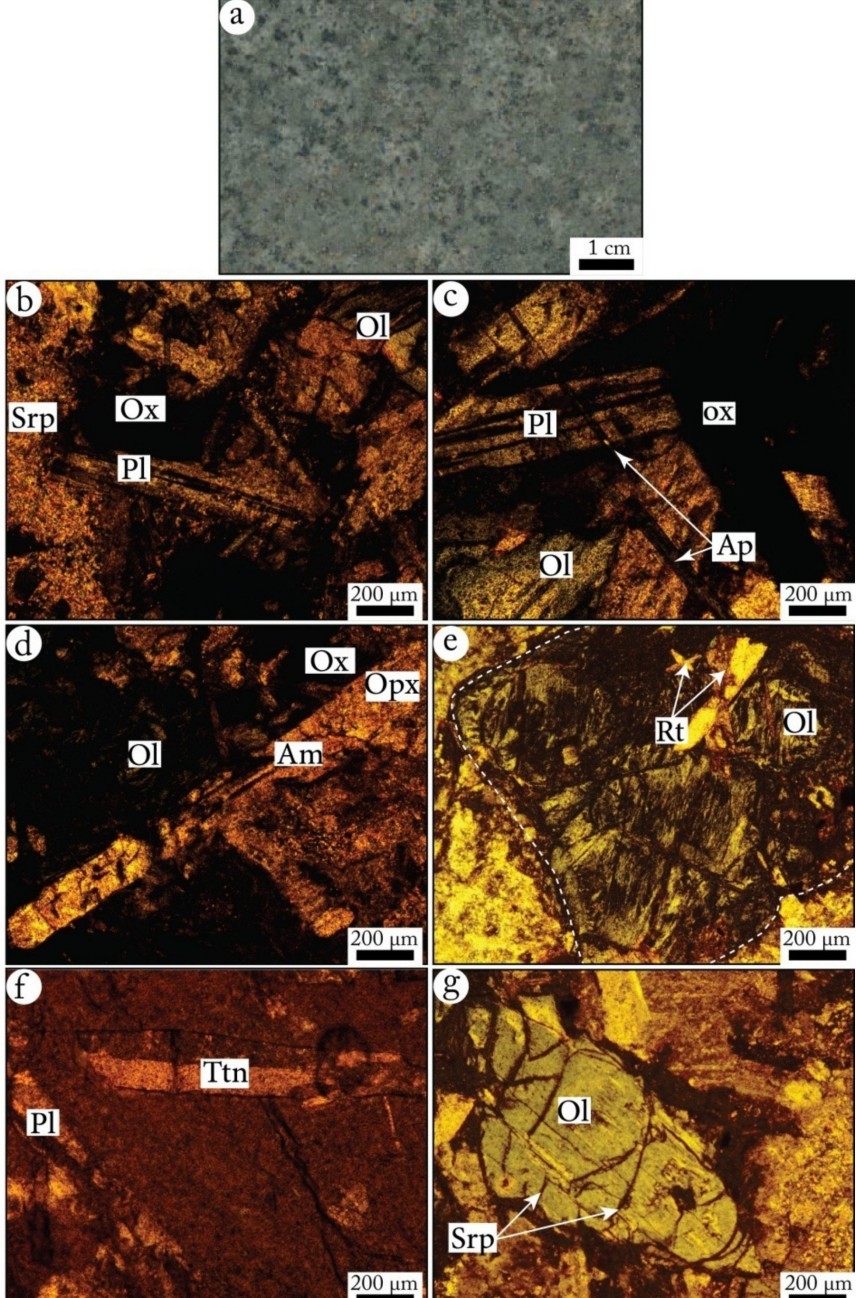

**Figure 10.** Samples of olivine-bearing gabbros. (**a**) Macroscopic view and (**b–g**): microphotographs of olivine-bearin gabbros (transmitted light-crossed nicols). Sericitized (Ser) plagioclases (Pl), amphibole (Am), orthopyroxene (Opx), serpentinized (Srp) olivine (Ol), Titanite (Ttn), apatite (Ap), rutile (Rt) and oxides (Ox).

The petro-mineralogical study shows that all these magmatic facies in the NNE-A underwent a moderate to important hydrothermal alteration. This is well marked on the pillow-lavas which are transformed into spilites by the intense albitization and carbonation of the plagioclases, and by the development of common chlorites.

The olivine-bearing gabbros have also been affected by an intense spilitization in which the plagioclases are sericitized and albitized, the pyroxenes are also amphibolitized and epidotized and the olivines are serpentinized and chloritized, whereas the dolerites show an alteration of plagioclases by sericite, and the pyroxenes are partially or totally amphibolitized (Figure 11).

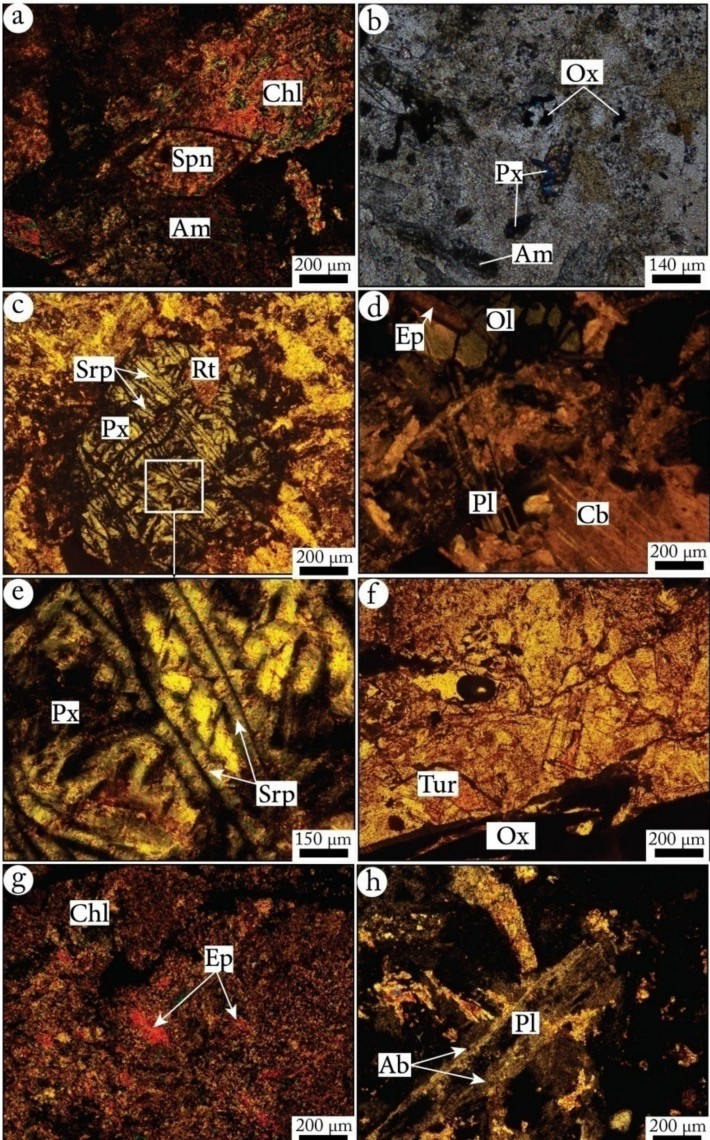

**Figure 11.** (**a**–**h**) Microphotographs of the alteration phases of the olivine-bearing gabbros: sericitized (Ser) and albitized (Ab) plagioclase (Pl), amphibolitized (Am) and epidotized (Ep) pyroxene (Px), serpentinized (Srp) and chloritized (Chl) olivine (Ol), tourmaline (Tur) and oxides (Ox) (transmitted light—crossed nicols).

The intensity of the hydrothermal alteration expressed by the replacement of the primary minerals and the development of albitization, sericitization and carbonation which affect theplagioclases, leads to an amphibolitization (development ofactinote), the and epidotization of pyroxenes, and chloritization followed by serpentinization of the olivines. The abundance of titaniferous minerals such assphene and rutile in close association with the aforementioned hydrothermal alteration confirms the presence of an evident ocean-floor hydrothermal metamorphism in this area [59–61].

*4.2. Mineralization*

4.2.1. Morphology

The Fe-Cu-Zn-Pb sulfide mineralization of the NNE-A sector is hosted either in sedimentary formations of Middle-Upper Visean and Upper-Namurian age, or in magmatic formations. This mineralization is disseminated or fills veinlets and veins (Figure 12).

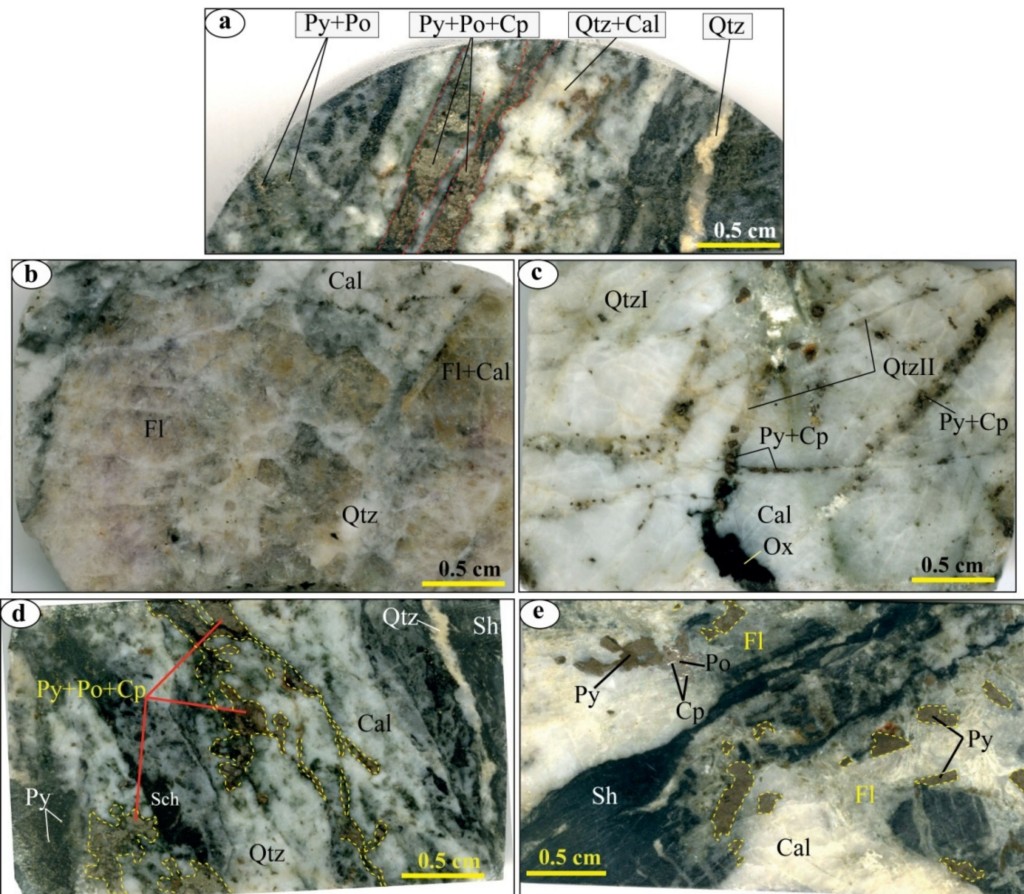

**Figure 12.** Textures of the ore bodies of NNE-A. (**a**,**d**) Lamination of pyrrhotite and pyrite in the host rock; (**b**) fluorite in association with calcite; (**c**) chalcopyrite, pyrite and iron oxides assemblage in vein; (**e**) disseminated chalcopyrite, pyrite, and pyrrhotite in shale, sandstone-carbonates and quartz. Po: pyrrhotite; Fl: Fluorite; Cal: Calcite; Cp: Chalcopyrite; Py: Pyrite; Ox: Oxides; Qtz: Quartz; Sh: Shale.

### 4.2.2. Mineralogy

The mineralization of the NNE-A area is associated with a gangue composed of fluorite, quartz, calcite and chlorite (Figure 13).

Microscopic observations combined with SEM and Electron Microprobe Analyses (EMPA), reveal the presence of diverse mineral assemblages composed of pyrite, pyrrhotite, marcasite, chalcopyrite, galena, sphalerite, glaucodot, ullmannite, hematite, magnetite and monazite.

On the basis of the textural relationships between different minerals, combined with their chemical study by using SEM and EMPA methods allowed us to define the paragenetic succession of the sulfide mineralization in NNE-A. This paragenetic sequence includes the following three main stages: (i) paragenesis I formed by pyrrhotite I, pyrite I, chalcopyrite I, sphalerite I, galena I, glaucodot, monazite, magnetite, ullmannite, and rutile associated with gangueof quartz, (ii) paragenesis II mainly constituted by fluorite, and (iii) paragenesis III composed by pyrrhotite II, pyrite II, chalcopyrite II, sphalerite II, galena II and hematite associated with a calcite gangue (Figures 14–16).

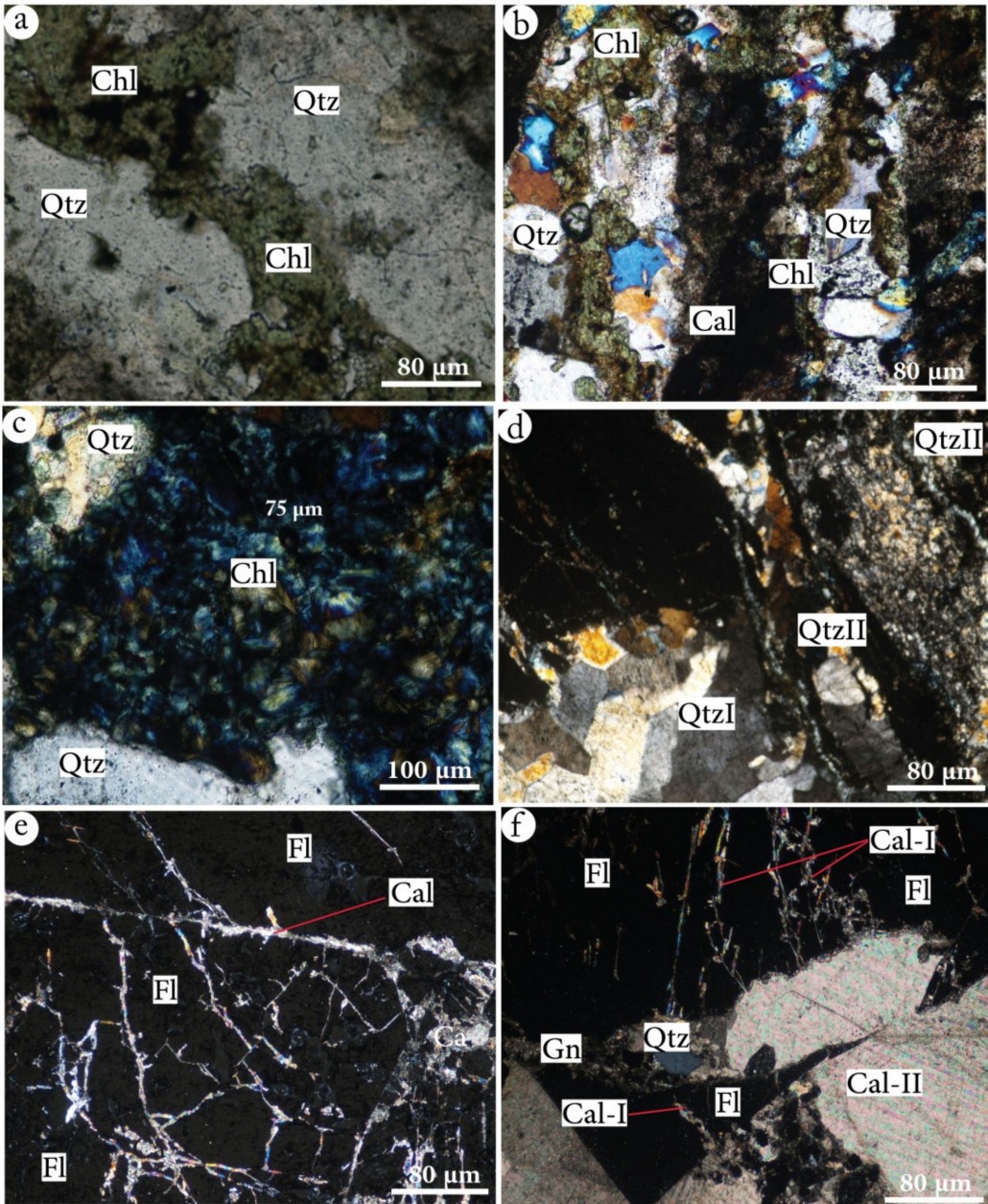

**Figure 13.** Microphotographs of different generations of quartz in association with chlorite, calcite, and fluorite (cross nicols, except for (**a**) where nicols are parallel). (**a**,**c**) Late veins of chlorites intersecting early veins of quartz; (**b**) calcite developed veins intersecting early quartz veins; (**d**) late quartz veins that crosscut the early veins; (**e**) late calcite veins intersecting fluorite; (**f**) calcite II developed in late veins and intersecting early veins of calcite I. Chl: Chlorite; Cal: Calcite; Fl: Fluorite; Qtz: quartz.

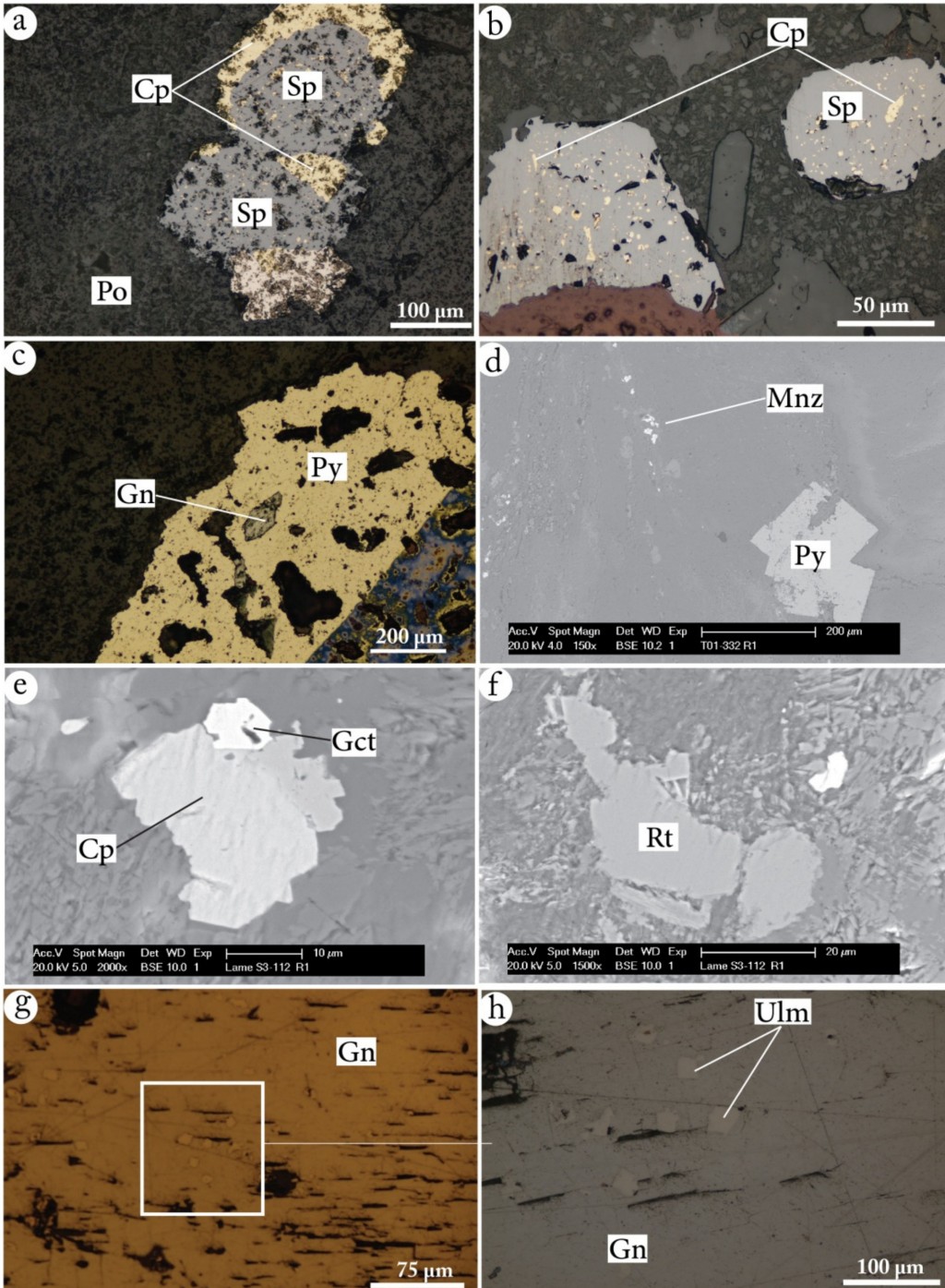

**Figure 14.** Microphotographs of ore minerals in gangue of quartz, calcite, and chlorite (reflected light; **a–c,g,h**) and back-scattered electrons (BSE; **d–f**). (**a**) Pyrrhotite and chalcopyrite coexisting with sphalerite in quartz-calcite-chlorite gangue; (**b**) sphalerite grains with inclusions of chalcopyrite in quartz-chlorite gangue; (**c**) pyrite minerals containing galena inclusions in a quartz-chlorite vein; (**d**) automorphic pyrite crystal sandstone; (**e,f**) association of chalcopyrite- glaucodot with rutile in dolerite; (**g,h**) inclusions of ullmannite crystals in galena. Po: pyrrhotite; Sp: Sphalerite; Cp: Chalcopyrite; Gn: Galena; Py: Pyrite; Gct: Glaucodot; Mnz: Monazite; Rt: Rutile; Ulm: Ullmannite.

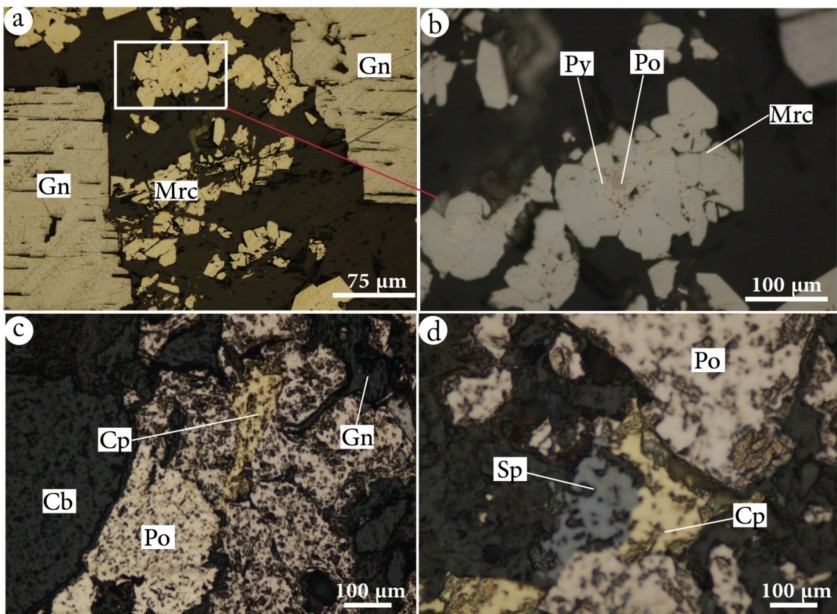

**Figure 15.** Microphotographs of ore minerals assemblage in the NNE-A (microscopic observation in reflected light; RL). (**a,b**) Pyrite and pyrrhotite relics in marcasite crystals, disseminated in a calcite vein that intersects galena; (**c,d**) pyrrhotite, chalcopyrite, galena and sphalerite in quartz-carbonate veins. Po: Pyrrhotite; Sp: Sphalerite; Cp: Chalcopyrite; Gn: Galena; Py: Pyrite; Mrc: Marcasite; Cb: Carbonates.

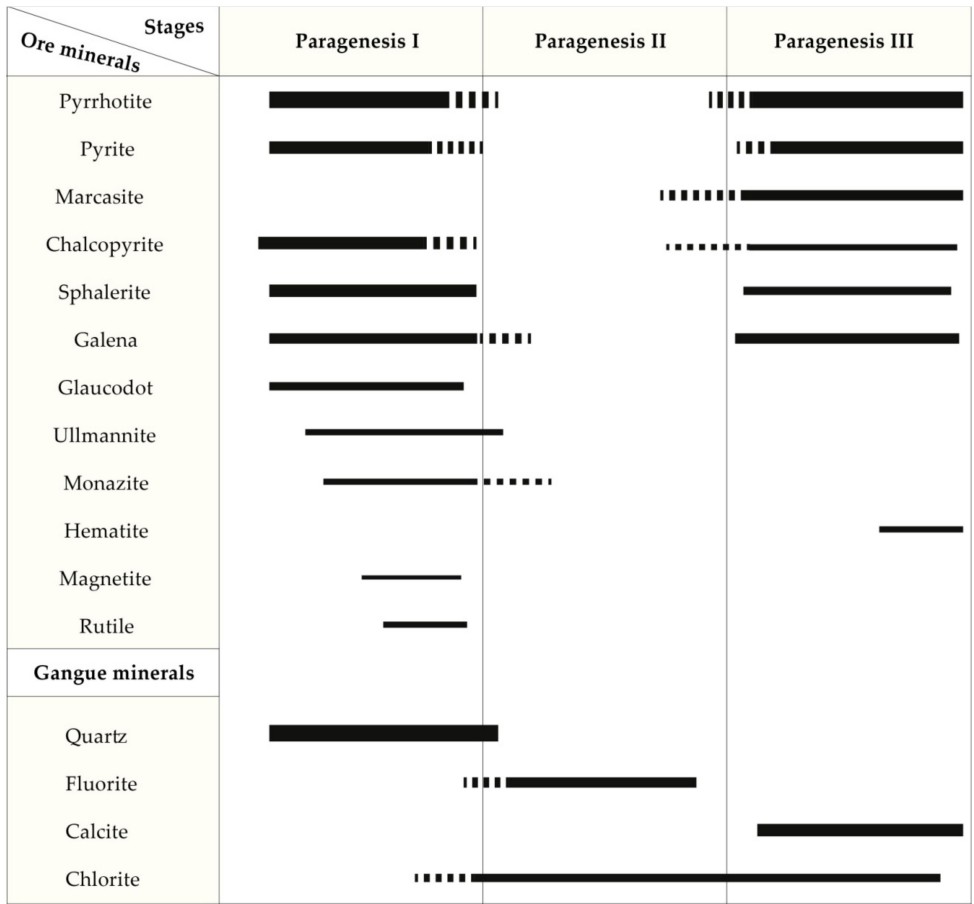

**Figure 16.** Mineral paragenetic succession of the NNE-A deposit.

### 4.2.3. Sulfide Chemistry

The chemical composition of all of the above-mentioned sulfide minerals were determined by conducting an electron microprobe analysis. The chemical analyses are provided in Table 1, and the main chemical features for the sulfide minerals are as follows:

**Table 1.** Electron microprobe analysis of NNE-A mineral paragenesis. (.): average amounts; [–]: minimum and maximum amounts.

| % Wt | Pyrite I | Pyrite II | Chalcopyrite | Pyrrhotite | Sphalerite I | Sphalerite II | Galena |
|---|---|---|---|---|---|---|---|
| Fe | (49.17) [48.83_ 49.54] | (49.23) [49.02_ 49.37] | (31.22) [30.68_ 31.73] | (61.82) [60.89_ 63.12] | (12.09) [11.00_ 14.42] | (9.99) [9.23_ 10.61] | (0.25) [0.04_ 0.46] |
| Zn | (0.02) [0.00_ 0.11] | (0.10) [0.03_ 0.14] | (0.08) [0.00_ 0.33] | (0.04) [0.00_ 0.19] | (53.69) [48.11_ 55.50] | (56.18) [55.70_ 56.54] | (0.10) [0.00_ 0.20] |
| S | (50.67) [49.70_ 50.98] | (49.66) [49.37_ 50.02] | (33.16) [32.50_ 33.55] | (37.74) [37.08_ 38.12] | (32.35) [32.12_ 32.61] | (32.15) [31.82_ 32.40] | (12.54) [12.41_ 12.67] |
| Pb | (0.20) [0.12_ 0.29] | (0.22) [0.20_ 0.24] | (0.14) [0.03_ 0.25] | (0.14) [0.04_ 0.27] | (0.08) [0.03_ 0.14] | (0.07) [0.03_ 0.11] | (86.57) [86.20_ 86.93] |
| Ag | (0.03) [0.00_ 0.06] | (0.03) [0.02_ 0.05] | (0.01) [0.00_ 0.04] | (0.02) [0.00_ 0.06] | (0.02) [0.00_ 0.04] | (0.01) [0.00_ 0.03] | (0.12) [0.05_ 0.19] |
| Cd | (0.03) [0.00_ 0.04] | (0.01) [0.00_ 0.01] | (0.01) [0.00_ 0.03] | (0.02) [0.00_ 0.04] | (0.30) [0.28_ 0.36] | (0.14) [0.05_ 0.38] | (0.01) [0.00_ 0.02] |
| In | (0.01) [0.01_ 0.03] | (0.05) [0.04_ 0.06] | (0.02) [0.00_ 0.05] | (0.01) [0.00_ 0.04] | (0.04) [0.00_ 0.05] | (0.03) [0.00_ 0.06] | (0.01) [0.00_ 0.02] |
| Sb | (0.03) [0.00_ 0.08] | (0.08) [0.04_ 0.11] | (0.01) [0.00_ 0.04] | (0.01) [0.00_ 0.04] | (0.01) [0.00_ 0.02] | (0.05) [0.00_ 0.10] | (0.02) [0.01_ 0.03] |
| Cu | (0.01) [0.00_ 0.03] | (0.03) [0.00_ 0.05] | (34.62) [34.05_ 34.93] | (0.01) [0.00_ 0.04] | (0.66) [0.03_ 3.83] | (0.10) [0.02_ 0.32] | (0.01) [0.00_ 0.02] |
| Ni | (0.01) [0.00_ 0.03] | (0.02) [0.02_ 0.03] | (0.00) [0.00_ 0.03] | (0.02) [0.00_ 0.07] | (0.01) [0.00_ 0.02] | (0.00) [0.00_ 0.01] | 0.00 [0.00_ 0.00] |
| Co | (0.01) [0.00_ 0.03] | (0.03) [0.03_ 0.04] | (0.00) [0.00_ 0.03] | (0.01) [0.00_ 0.06] | (0.01) [0.00_ 0.03] | (0.03) [0.00_ 0.05] | (0.02) [0.01_ 0.03] |
| As | (0.04) [0.00_ 0.10] | (0.01) [0.00_ 0.03] | (0.02) [0.00_ 0.07] | (0.02) [0.00_ 0.06] | (0.04) [0.00_ 0.06] | (0.02) [0.00_ 0.04] | (0.02) [0.01_ 0.02] |
| Total | (100.21) [99.17_ 100.78] | (99.47) [99.15_ 99.69] | (99.29) [98.67_ 99.97] | (99.86) [98.60_ 101.20] | (99.28) [98.37_ 100.49] | (98.77) [98.43_ 99.77] | (99.66) [99.14_ 100.18] |
| **% Atomic** | | | | | | | |
| Fe | (35.69) [35.47_ 35.91] | (37.33) [36.15_ 39.58] | (26.11) [25.84_ 26.44] | (48.41) [48.06_ 49.21] | (10.50) [9.65_ 12.44] | (8.75) [8.09_ 9.32] | (0.54) [0.08_ 1.00] |
| Zn | (0.03) [0.00_ 0.09] | (0.02) [0.00_ 0.07] | (0.06) [0.00_ 0.24] | (0.03) [0.00_ 0.12] | (39.84) [35.45_ 40.96] | (42.01) [41.75_ 42.32] | (0.19) [0.00_ 0.38] |
| S | (64.16) [63.82_ 64.41] | (62.51) [60.32_ 63.73] | (48.31) [47.82_ 48.89] | (51.47) [50.68_ 51.82] | (48.94) [48.53_ 49.16] | (49.02) [48.70_ 49.45] | (47.87) [47.80_ 47.94] |
| Pb | (0.04) [0.03_ 0.06] | (0.03) [0.02_ 0.04] | (0.03) [0.01_ 0.06] | (0.03) [0.01_ 0.06] | (0.02) [0.01_ 0.03] | (0.02) [0.01_ 0.03] | (51.13) [50.88_ 51.39] |
| Ag | (0.01) [0.00_ 0.02] | (0.01) [0.007_ 0.01] | (0.01) [0.00_ 0.02] | (0.01) [0.00_ 0.02] | (0.01) [0.00_ 0.02] | (0.00) [0.00_ 0.01] | (0.14) [0.05_ 0.22] |
| Cd | (0.01) [0.00_ 0.01] | (0.01) [0.00_ 0.01] | (0.01) [0.00_ 0.01] | (0.01) [0.00_ 0.02] | (0.13) [0.12_ 0.15] | (0.06) [0.02_ 0.16] | (0.01) [0.00_ 0.02] |
| In | (0.01) [0.00_ 0.02] | (0.01) [0.01_ 0.02] | (0.01) [0.00_ 0.02] | (0.01) [0.00_ 0.01] | (0.02) [0.00_ 0.02] | (0.01) [0.00_ 0.02] | (0.01) [0.00_ 0.02] |
| Sb | (0.01) [0.00_ 0.03] | (0.03) [0.02_ 0.04] | (0.00) [0.00_ 0.01] | (0.00) [0.00_ 0.01] | (0.00) [0.00_ 0.01] | (0.02) [0.00_ 0.04] | (0.02) [0.01_ 0.03] |
| Cu | (0.01) [0.00_ 0.02] | (0.01) [0.00_ 0.03] | (25.45) [25.04_ 25.76] | (0.01) [0.00_ 0.03] | (0.50) [0.02_ 2.90] | (0.08) [0.01_ 0.25] | (0.02) [0.00_ 0.05] |
| Ni | (0.01) [0.00_ 0.02] | (0.01) [0.00_ 0.02] | (0.00) [0.00_ 0.02] | (0.02) [0.00_ 0.05] | (0.00) [0.00_ 0.02] | (0.00) [0.00_ 0.01] | 0.00 [0.00_ 0.00] |
| Co | (0.01) [0.00_ 0.02] | (0.01) [0.00_ 0.02] | (0.00) [0.00_ 0.02] | (0.01) [0.00_ 0.04] | (0.01) [0.00_ 0.03] | (0.02) [0.00_ 0.04] | (0.05) [0.03_ 0.07] |
| As | (0.02) [0.00_ 0.05] | (0.01) [0.00_ 0.01] | (0.01) [0.00_ 0.04] | (0.01) [0.00_ 0.03] | (0.02) [0.00_ 0.04] | (0.01) [0.00_ 0.03] | (0.03) [0.02_ 0.03] |
| Total | (100.00) [99.95_ 100.03] | (99.99) [99.97_ 100.01] | (100.00) [99.999_ 100.00] | (100.00) [99.999_ 100.002] | (100.00) [99.999_ 100.001] | (100.00) [100.00_ 100.001] | (100.00) [100.00_ 100.001] |

Pyrrhotiteis found in close association with sphalerite, chalcopyrite and galena, and hosted by quartz-carbonate-chlorite veins. The results, presented in Table 1, show two

chemical compositions that are slightly different for this sulfide mineral, which were already determined on the basis of the polarizing microscopy as pyrrhotite I and pyrrhotite II (Figure 17a).

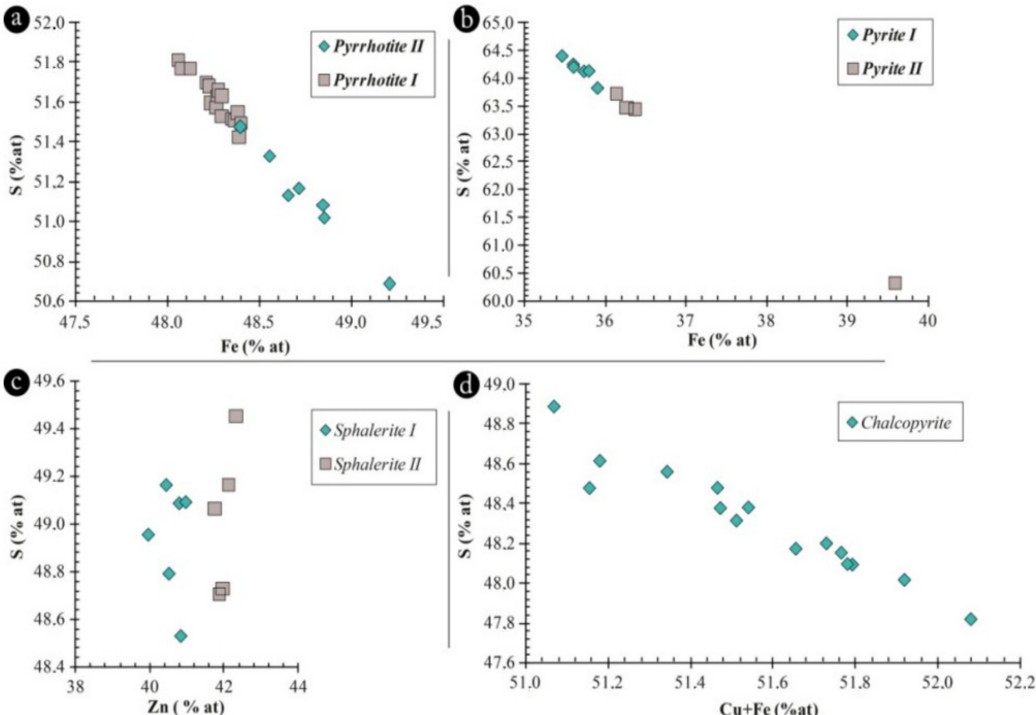

**Figure 17.** Geochemical data (in %at) for sulfide mineralization of NNE-A. Correlation diagram between Fe and S of pyrrhotites (**a**), between Fe and S in pyrites (**b**), between Zn and S in sphalerites (**c**), between Cu + Fe and S in chalcopyrite (**d**).

The EMPA shows two distinct chemical compositions of pyrrhotite. The Fe content ranges from 48.06 to 49.21at.% Fe, with an average value of 48.41at.% Fe, whereastheScontents are between 50.68 and 51.82at.% S with an average value of 51.47at.% S. The Fe/S atomic ratio ranges between 0.92 and 0.97. The structural formula of pyrrhotite is therefore, $Fe_{0.96}S_{1.03}$.

Pyrite: the microscopic investigations allowed us to distinguish two generations of pyrite. The results of the chemical analysis are presented in Table 1 below. Microprobe analyses of pyrites reveal that they experience constant composition independently of the inferred generation. Furthermore, the Fe and S contents range from 36.15 to 39.58at.% Fe and 60.32 to 63.73at.% S, with average values of 37.35at.% Fe and 62.51at.% S, respectively. The punctual analysis of pyrites shows a composition close to the known theoretical structural formula of $FeS_2$. The average structural formula of the analyzed pyrites from the NNE-A sector was $Fe_{1.08}S_{1.91}$ (Figure 17b).

Sphalerite: The chemical composition of the two generations of sphalerite is given in Table 1. The EMPA of the early sphalerite indicates a Zn, S, and Fe contents ranging from 35.45 to 40.96at.% Zn, 48.53 to 49.16at.% S and 9.65 to 12.44at.% Fe, with average compositions of 39.84at.% Zn 48.94at.% S and 10.50at.% Fe, respectively. Sphalerite, meanwhile, exhibits a sulfur content similar to the early sphalerite (49.06 to 49.45at.% S), with slight changes in Zn and Fe contents (41.75 to 42.32at.% Zn and 8.09 to 9.32at.% Fe, respectively, with average values of 42.01at.% Zn and 8.75at.% Fe) (Table 1).

On the Zn vs. Fe and Zn vs. S diagrams, we demonstrate that late sphalerite is less rich in Fe (9.32at.%) than the early one (14.41at.%) which ispoorerin Zn. So, Fe is the main element that omorphically replaces Znin the sphalerite's structure and has a negative correlation. It is noted that the variations in the Fe/Zn ratio (0.24 to 0.27) indicate a slight excess of Fein early sphalerite. The sand Zn contents show a positive correlation; these

two chemical elements are relatively low in early sphalerite compared to late sphalerite (Figure 17c). The calculated structural formulas for the two sphalerites are then as follows:

Early sphalerite: $Zn_{0.80}Fe_{0.21}S_{0.98}$

Late sphalerite: $Zn_{0.84}Fe_{0.17}S_{0.98}$

In the NNE-A sector, we note coexistence in the same paragenesis of sphalerite I, pyrrhotite I, and chalcopyrite, which suggests that these mineral phases are in equilibrium. This equilibrium is used to estimate the temperatures of sphalerite formation. The molar contents of Fe and S in sphalerite are as follows: 16.34 mole % and 25.37 mole %, respectively. Using the FeS-ZnS equilibrium diagram of [62] (Figure 18), the projection of Fe and S values reveals that the earlier sphalerite was formed at about 420 °C. This temperature seems to be in line with the work of [63], which indicated that the presence of chalcopyrite blebs in sphalerite would signify a crystallization temperature between 200 and 400 °C. The ate sphalerite, meanwhile, has a temperature of formation close to 324 °C.

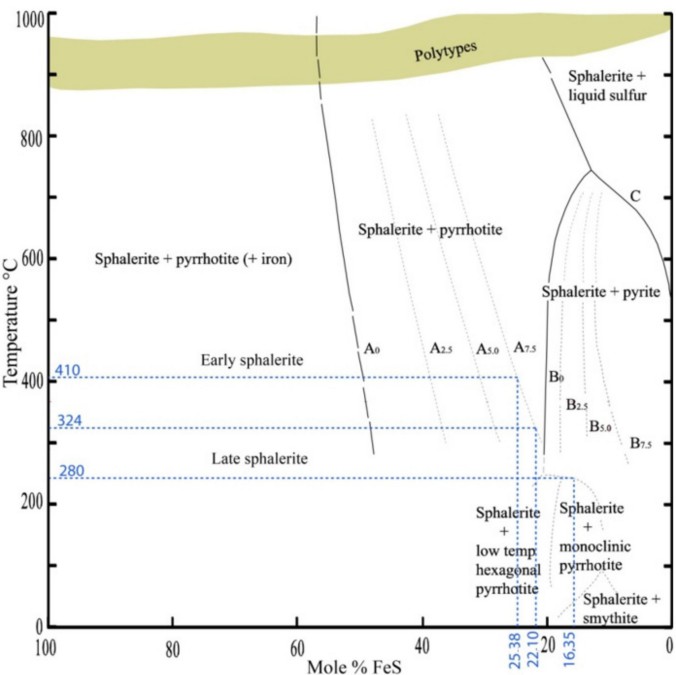

**Figure 18.** Estimation of the temperature conditions of NNE-A sphalerites deposition from the phase diagram of the FeS-ZnS system [64], from the temperature (°C) versus FeS (mole %) diagram.

Similarly to sphalerite, analyzed chalcopyrite was sampled from the veins filled with chlorite-quartz, calcite-quartz, and quartz or calcite gangue minerals. This chalcopyrite is associated with pyrrhotite, sphalerite and galena. The chemical compositions of chalcopyrite are presented in Table 1. The analyses of chalcopyrite with an electron microprobe show almost a stoichiometric composition of chalcopyrite, with a composition close to the theoretical formula of $CuFeS_2$. The compositions of the major elements vary slightly: 25.04 to 25.76at.% Cu; 25.84 to 26.44at.% Fe and 47.82 to 48.89at.% S. Chalcopyrite has similar proportions of Fe, with an average composition of 26.11at.% Fe. The Fe vs. Cu/Fe diagram shows a clear negative correlation, while the Fe vs. Cu/S one indicates a positive correlation. Indeed, variations in the Cu/Fe ratio in the chalcopyrite are interpreted as the result of its rebalancing with other accompanying minerals during metamorphic or hydrothermal events [64]. The S contents vary slightly, with an average composition of 48.31at.% S, indicating a slight enrichment in the Scontent in the analyzed chalcopyrite. These variations in the chemical composition of chalcopyrite could reflect the substitutions of its major elements (Cu, Fe, and S) with some trace elements (Pb, Zn, Cd, In, Sb, As . . . ) present in this mineral with non-negligible contents. The average structural formula of the analyzed chalcopyrite is $Cu_{1.01}Fe_{1.04}S_{1.93}$ (Figure 17d).

### 4.3. Fluid Inclusions Results

The petrographic and microthermometric studies of the fluid inclusions, identified in quartz, calcite and fluorite, show that they are frequently disseminated in the mineral matrix or align along growth planes of the host mineral (quartz, calcite and fluorite) (Figure 20). Fluid inclusions are therefore qualified as primary, or when trailing in planes crossing the crystals boundaries they are considered as secondary. The characteristics of the fluid inclusions (host-mineral, arrangement, chronology, thermometry, nature, composition and homogenization temperature), summarized in Figure 19, allowed us to distinguish the following:

| Sample reference | Host mineral | Numbre | Chronology | Type | Morphological properties | | | Microthermometric caracteristic | | | | Paragenetic stages |
|---|---|---|---|---|---|---|---|---|---|---|---|---|
| | | | | | Form | Size (µm) | Rv (%) | Te (°C) | Tmi (°C) | Salinity (Wt%. eq.NaCl) | Th (°C) | |
| TS10 | Quartz | 116 | primary to pseudo-secondary | L1 | regular rounded irregular | 10 to 30 | < 45 | -75 to -70 | (-15; -5) | 20 to 25 | (550;600) | pyrrhotite I pyrite I chalcopyrite I sphalerite I galena I glaucodot monazite hematite magnetite rutile ullmannite |
| | | | | L2 | regular rounded irregular | 10 to 30 | < 45 | -75 to -70 | (-15;-5) | 10 to 20 | [250;550] (350;400) | |
| | | | | V | regular | 10 to 20 | ≥ 50 | -75 to -60 | (-15;-5) | 10 to 20 | [250;550] (450;550) | |
| TS2 | Fluorite | 51 | Primary | $S_f$ | regular rounded triangular irregular | 15 to 30 | 10 to 15 | -70 to -85 | [-15;-35] (-15;-25) | 30 to 35 | [120;200] (120;160) | – |
| | | | Secondary | $L_f$ | regular rounded triangular irregular | 10 to 30 | 5 to 40 | -80 to -70 | [-5;-20] (-10;-20) | 10 to 20 | [120;280] (160;200) | |
| TS7 | Calcite | 48 | Primary | $L_{Ca}$ | regular irregular | 5 to 30 | 10 to 40 | -75 to -80 | [-5;-20] (-5;-15) | 10 to 20 | [200;350] (200;250) | pyrrhotite II pyrite II chalcopyrite II sphalerite II galena II |

**Figure 19.** Summary of the fluid inclusions of the NNE-A deposit. Rv: Ratio vapour, Te: Eutectic, Tmi: melting temperature of the ice.

(1) In quartz, two types of fluid inclusions (L, and V-type) were identified (Figure 20B). They initially corresponded to an early liquid aqueous fluid of high temperature (HT) (Th = 550–600 °C and Wt% 20–25eq.NaCl, type L1), evolving, by mixing with a less salty and colder external fluid, towards two aqueous fluids (L2 and V). These fluids have very similar thermometric characteristics with more moderate homogenization temperatures (Th= 300–450 °C) and low-salinities: 10–20Wt % eq. NaCl (Figure 21B (b)). The eutectic temperatures ($T_e$) measured, show values between −75 and −70 °C (Figure 20B (a)). These temperatures, generally lower than the eutectic temperature of the simple binary system H2O-NaCl (Te =−21.1 °C), suggest the presence of salts other than NaCl in the fluid. The values obtained are close to the eutectic of the H2O-NaCl-Li system (Te = − 75 °C). This characteristic supports that the different types of fluids distinguished in the quartz derive from the evolution of the same aqueous fluid of HT and high salinity (L1). This evolution consists of the dilution of the magmatic fluid (type L1), during its ascent to the surface, by a meteoric fluid of low temperature (LT) and low salinity, giving rise to the aqueous fluids L2 and V (Figure 22).

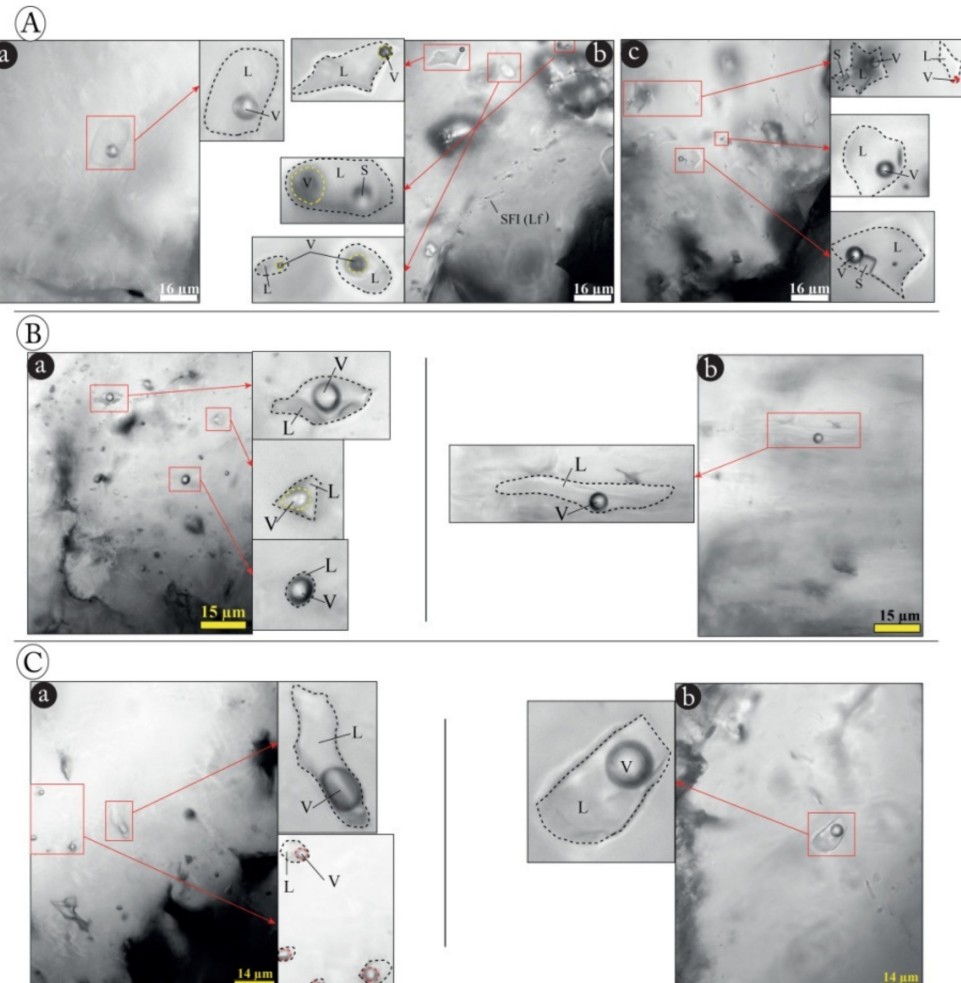

**Figure 20.** Photomicrographs of typical fluid inclusions in quartz, fluorite, and calcite. (**A** (a,b,c)) Secondary biphasic fluid (L and V) L-type inclusions and primary fluid (L, V, and S) S-type encountered in fluorite in plane-polarized light (PPL); (**B** (a,b))biphasic fluid (L and V) L-type and V-type inclusions encountered in quartz (PPL); (**C** (a,b)) biphasic fluid (L and V) L-type inclusions encountered in calcite (PPL).L: Liquid; V: Vapor; S: Solid; SFI (Lf): secondary fluid inclusions of the fluorite.

(2) In fluorite, the following two fluids are distinguished: (i) type Sf corresponding to a hypersaline fluid; characterized by a homogenization temperature between 120 and 160 °C and a salinity ranging from 30 to 35 Wt% eq. NaCl, and (ii) type Lf consisting of a moderately salty aqueous fluid with a homogenization temperature between 150 and 200 °C, and a salinity between 10 and 20 Wt % eq. NaCl, (Figure 21A (a,b)), and (iii). The Th vs. Tfs diagram (Figure 21A (c)), shows that the total homogenization of all the inclusions of this type occurred in the liquid phase due to the disappearance of the vapor after that of the solid (S). This reflects the homogeneity of the initial fluid trapped and the son character of salt crystals.

(3) In calcite, only an aqueous fluid is recognized (type LCa). It is characterized by a relatively low temperature (Th between 200 and 250 °C), a rather moderate salinity (between 10 and 20 Wt % eq. NaCl) (Figure 21C (b,c)) and eutectic temperatures ranging from −60 °C to −95 °C with two modes (−65 to −70 °C and −75 to −80 °C) (Figure 21C (a)).

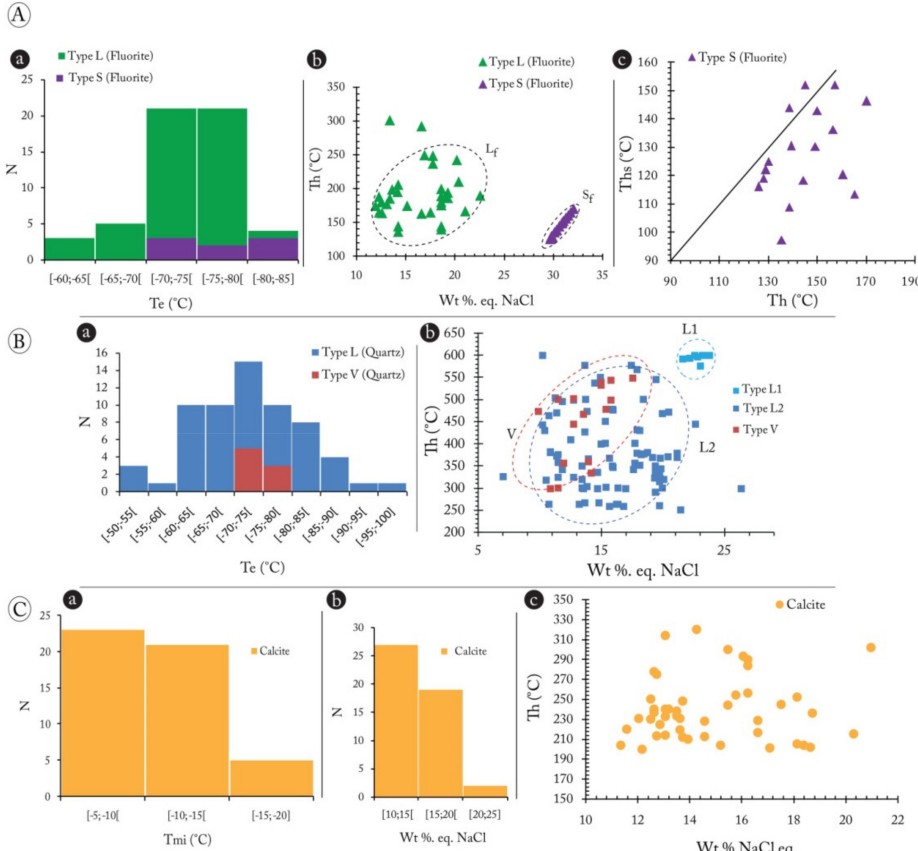

**Figure 21.** Analyses of different types of fluid inclusions in quartz, fluorite and calcite. (**A** (a)) Histograms of eutectic temperatures of fluid inclusions for L and S-type in fluorite; (**A** (b)) salinity scatter plot—homogenization temperatures of L-type fluid inclusions offluorite; (**A** (c)) scatter plot temperatures of homogenizations (SHt)-(Ht) of S-type fluid inclusions of fluorite. (**B** (a)) Histogram of the homogenization temperatures (Ht) of fluid inclusions of quartz; (**B** (b)) salinity scatter plot–homogenization temperature of L and V-type fluid inclusions of quartz. (**C** (a)) Histogram of ice-melting temperatures (IMT); (**C** (b)) salinity of L-type fluid inclusions of calcite; (**C** (c)) salinity scatter plot– homogenization temperatures of L-type fluid inclusions of calcite. L: Liquid; V: Vapor; S: Solid.

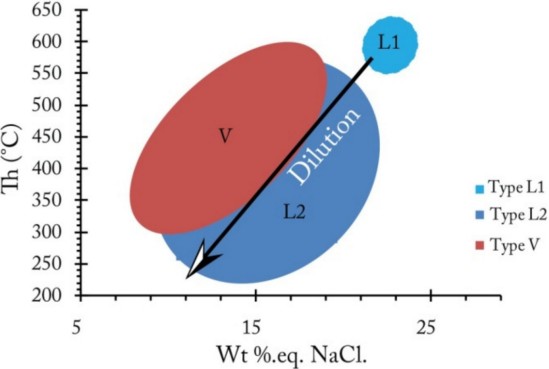

**Figure 22.** Salinity vs. homogenization temperature scatters plot of fluid inclusions shows the evolution of the salty aqueous fluid of quartz.

The homogenization temperatures and salinities of the aqueous fluid inclusions of fluorite ($L_f$), show great similarity with those measured in calcite ($L_{Ca}$) (Figure 23). The calcite crystals were formed after the deposition of the fluorite ones, as shown by the petrographic study (Figure 13e,f). Therefore, this type of fluid inclusions would correspond to the aqueous fluid ($L_{Ca}$ of calcite) that would have circulated in the fluorite micro-vein.

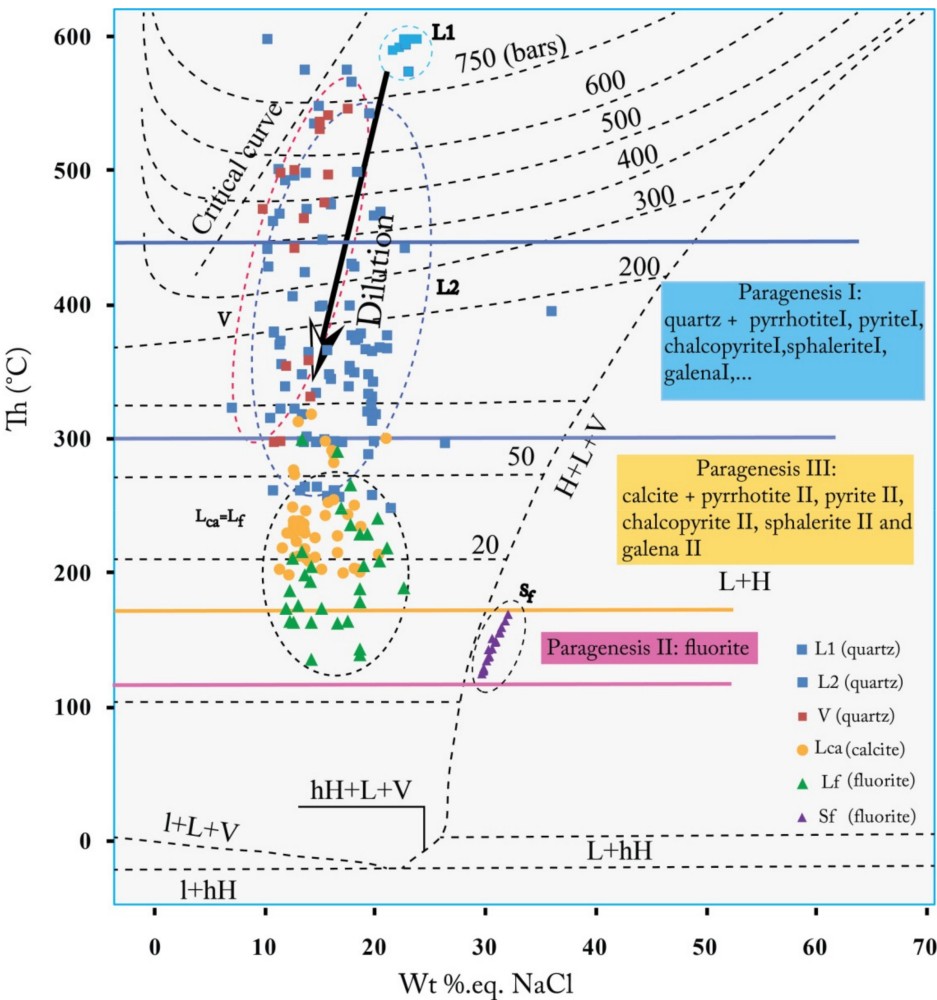

**Figure 23.** Salinity vs. homogenization temperature dispersion diagram with isobar sections of fluid inclusions of type L, V, and S encountered in the NNE-A sector. L: Liquid; V: Vapor; S: Solid.

The study of the fluid phases revealed three distinct fluids, which are as follows:

(i) An aqueous fluid of high temperature and high salinity (L1), associated with quartz and with a magmatic origin. This fluid would have mixed, during its ascent to the surface, with a fluid of marine origin percolating through the faults and fractures that affected the oceanic crust at depth. This mixture, materialized by the L2 and V fluids of lower temperature and lower salinity, would be at the origin of the precipitation of the metallic charge of these fluids and thus the deposition of the primary sulfide paragenesis around 300 to 450 °C; (ii) a low-temperature hypersaline fluid (brine), specific to fluorite, whose characteristics are identical to those of brines in the fluorite of the El Hammam vein deposit [34,65,66] and (iii) a fluid of medium salinity and moderate temperature $L_{Ca}$, characterizing calcite and whose traces are found, in secondary inclusions, in fluorite ($L_f$) and in quartz, thus confirming the posteriority of calcite compared to the other minerals (quartz and fluorite). Fluorite Lf and calcite LCa fluids are associated with the late sulfide paragenesis that also occurs in the fluorite mineralization of the El Hammam district deposits.

## 5. Interpretation and Discussion

Before discussing the petrological and geodynamic significance of magmatic rocks, it is necessary to redraw some of the evidence: (i) the three magmatic facies recognized are the major constituents of an oceanic crust (gabbros at the base, dolerite dykes in the medium (sheeted dikes) and basaltic lava flows at the top). The upper unit is often covered, at the level of the ocean floor, by sediments; (ii) the volcanic facies of these rocks correspond to a

basalt exposed as pillow structures which could testify to their deep-sea deposition; (iii) the transformations that affected the primary minerals (olivines, pyroxenes and plagioclases) of the three magmatic facies are the metamorphism (green schist facies) and ocean-floor hydrothermal alteration (serpentinization, spilitization, chloritization, epidotization, etc.); (iv) the geochemical affinity of these rocks is that of an alkaline magmatism with a tholeiitic differentiation trend [29].

It should be assumed that the petrological features and the geochemical signature of the rocks mentioned above indicate an oceanic crust resulting from the opening of an intracontinental rift due to an extensive tectonic regime. This rifting history was marked by the reactivation of normal faults; the thinning of the crust, and the collapsing blocks from the crust. These collapses are coupled with an isostatic ascent of the oceanic crust, and the underlying mantle leading to its partial fusion. The basaltic magma, resulting from this fusion, spreads at the level of the oceanic floor, which begins to form under a small slice of water (beginning of oceanization).Hence, the basaltic pillow lavas were formed by the eruption of a volcanic magma under oceanic water. The fracturing of the continental crust during the extension allowed for the infiltration of sea water in an oceanic crust, whose main components (basalts and gabbros) are very poor in water (less than 1%). It puts the main mineralogical assemblages of these rocks (olivines, pyroxenes and plagioclases) in a very strong chemical imbalance with their new environment. Moreover, the mantle is never far, the range of temperatures in which this circulation of water takes place is still very high, and thus the reactions of the rock's re-equilibration is favored. The water provides the OH- necessary inthe production of the hydrated phases of low-temperatures such as serpentine minerals (serpentinization) which are developed in gabbros resulting from the hydration of the olivines. However, water also carries a large quantity of dissolved salts, in particular NaCl and chlorine, which are added to hydrogen to decrease the pH and increase the aggressivity of water. Sodium, exchangeable with the calcium of plagioclases or basalt's pyroxenes, leads to the transformation of the basalts into spilites (spilitization). The basic idea is that the primary minerals (pyroxenes and calcium feldspars) of the basalts will undergo an alteration during the underwater lava effusion due to their interaction with salty water. In particular, the aforementioned plagioclase feldspars, an integral part of basalt, give rise to a hydrothermal metamorphism, which causes them to evolve from possessing a content rather rich in calcium to a rather high content of sodium (from sodium ions in solution in seawater).The continuous feldspathic-plagioclase solid solution moves thus from the calcium pole (anorthite) to the sodium pole (albite, albitization).In addition, the calcium released during substitution with sodium, combines with the $CO_2$ present in the fluid, and precipitates in the form of calcite (carbonation), giving rise to the characteristic white pustules of spilites. At the outcrop, these whitish pustules practically disappear, leaving place for vacuoles. During hydrothermal circulation, the fluid percolates into the oceanic crust, heats and loads metals, such as Fe, Mn, Mg, Cu, Zn, P, and sulfur and silicon produced by the following alteration reactions:

$$Mg_2SiO_4 \text{ (olivine)} + 2CO_2 = 2MgCO_3 \text{ (magnesite)} + SiO_2 \text{ (quartz)}$$

$$Mg_2Si_2O_6 \text{ (pyroxene)} + 2CO_2 = 2MgCO_3 \text{ (magnesite)} + 2SiO_2 \text{ (quartz)}$$

As it rises to the surface, the L1 fluid mixes with external fluids. It cools down and changes its physico-chemical properties (decrease in temperature, decrease in salinity, increase in fO2, . . . ) and evolves towards the L2 and V fluids which end up depositing the primary sulfide mineral paragenesis (pyrrhotite I, pyrite I, chalcopyrite I, sphalerite I, galena I, ullmannite, glaucodot, monazite, hematite, magnetite and rutile) associated with quartz (Figure 24). Thermometric estimates made either from the study of fluid phases or from mineralogical assemblies (sphalerite-pyrrhotite) indicate a range of deposition temperatures of the primary sulfide mineralization of between 300 °C and 450 °C. Later, the arrival in the system of a second hypersaline ($S_f$: 25 to 35 Wt% eq. NaCl) and low-temperature fluid (120 to 160 °C), provide the origins of the fluorite deposit. The circulation

of a third fluid that is aqueous, moderately salty (10 to 20 Wt% eq. NaCl) and fairly hot (200 °C to 250 °C), associated with calcite ($L_{Ca}$), and found in secondary expression in fluorite and quartz, is responsible for the deposition of late sulfide paragenesis (pyrrhotite II, pyrite II, chalcopyrite II, sphalerite II, hematite). Fluorite brines ($S_f$) and aqueous fluid ($L_{Ca}$) calcite in the sulfide mineralization of this sector have the same thermometric characteristics as those described in the fluorite fluids and paragenesis, with the calcite of late sulfides filling the veins of the El Hammam deposit [36,65–67].

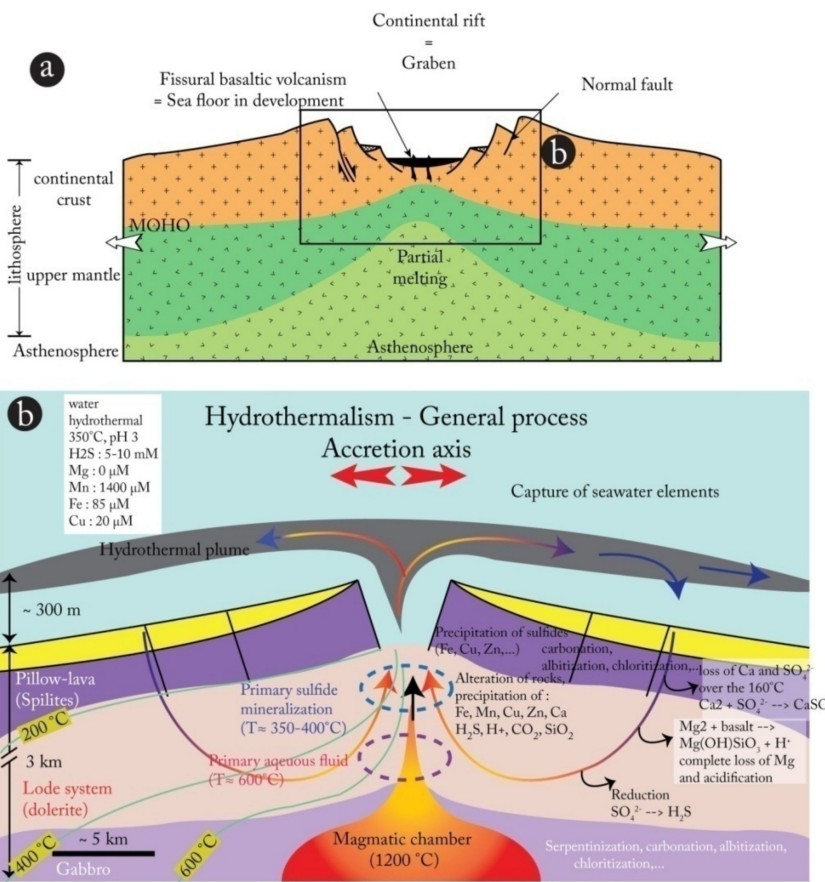

**Figure 24.** Proposed genetic implementation model for primary sulfide mineralization at the NNE-A sector (**a**,**b**) [5].

It was deduced that the deposit of sulfide and fluorite ore mineralization in the NNE-A sector is, therefore, the result of the succession of three mineralizing events (Figure 24): (a) an early, high-temperature hydrothermal event related to the placement of basic rocks in the NNE-A sector, with which the primary sulfide paragenesis is associated. In addition to the geological and petrogenic arguments developed above, we can cite others, which support this hypothesis and exclude the connection of this event to the felsic magmatism of El Hammam: (1) high temperatures (500–600 °C) of the L1fluid, although they are detected in the magmatic fluids of El Hammam (fluids of the stanniferous stage of the skarn at Sn-W – B of El Hammam [33,68], are not sufficient to reattach this fluid of HT (L1) at the stanniferous stage, for two reasons: (i) the nature of the L1 fluid, which is less rich in salts and has a lower eutectic than the stanniferous fluid [68]; (ii) the mineral paragenesis resulting from the L1fluid does not contain any trace of Sn, W, or B which characterize the mineralization associated with the perigranitic magmatism of El Hammam, and whose precipitation is compatible with HT. (2) The HT (300–450 °C) estimated for the precipitation of primary sulfide paragenesis, either from mineralogical assemblages (pyrrhotite-sphalerite, Fe-rich sphalerite I and abundance of chalcopyrite inclusions in sphalerite), or on the basis of the fluid inclusions study, is characterized by the sea floor environment. (3) The ore deposit

contains numerous Ni and Co minerals including ullmannite and glaucodot which have an intimate relationship with mafic and ultramafic rocks. The typical example of this mineralogical relationship and geochemical affinity of the Co and Ni mineralization with the mafic and ultramafic formations have been mainly described at the Bou Azzer Co-Ni-Fe arsenide deposits [66]. These are all arguments for associating this early hydrothermal event with the mafic magmatism of this sector. (b) A second low-temperature hydrothermal event associated with fluorite, and (c) a third, later event that causes late sulfide paragenesis deposition and its associated carbonation. These last two hydrothermal events are linked either to the felsic magmatism of El Hammam, as is generally claimed by many authors studying the fluorite and sulfides of El Hammam [34,65,66], or to the effect of the thermal flux generated by the basic volcanism in connection with the Triassic-Jurassic opening of the Atlantic Ocean proposed by [69] following the dating of the fluorite mineralization of El Hammam using the $^{40}$Ar/$^{39}$Ar method on potassic feldspar.

## 6. Conclusions

The results obtained in this work allowed us to conclude that:

i.    The following two Paleozoic groups compose the NNE-A region: (1) a meta-sedimentary group comprising two units; the first one is made of limestone and schisto-sandstone of middle Visean, and the second one is made of flyschoid of the Upper Visean-Namurian, and (2) a magmatic group, described here for the first time, materialized by volcanic (pillow-lavas), hypo-volcanic (dolerites) and plutonic (olivine-bearing gabbros) rocks;

ii.   Pillow basaltic lavas and olivine-bearing gabbros have alkaline affinity, while dolerites have transitional alkaline affinity (alkaline-tholeiitic). They suggest the setting up of this mafic magmatism in intracontinental rifting, beginning with oceanization. These basic rocks have undergone an oceanic hydrothermal alteration, causing their serpentinization, spilitization, chloritization and carbonation;

iii.  Vein or disseminated Fe-Cu-Zn-Pb sulfide and fluorite mineralization, hosted either in Visean sedimentary formations or in the magmatic ones, were established according to the following three paragenetic stages: (i) paragenesis I, with quartz gangue, composed of pyrrhotite I, pyrite I, chalcopyrite I, sphalerite I, galena I, glaucodot, ullmannite, magnetite and rutile, (ii) paragenesis II, mainly formed by fluorite and (iii) paragenesis III, with pyrrhotite II, pyrite II, chalcopyrite II, sphalerite II, galena II and hematite with a carbonate gangue;

iv.   Equilibrium sphalerite-pyrrhotite reveals that the earlier sphalerite was formed at about 420 °C and the late sphalerite at 320 °C. These temperatures were confirmed by the fluid inclusions study;

v.    The deposition of sulfide and fluorite mineralization is characterized by the circulation of the following three fluids: (i) an aqueous and salty fluid of HT (L1: Th = 550–600 °C and $X_{NaCl}$ = 20–25 Wt% eq. NaCl), resulting from the circulation of magmatic fluids in depth during the rifting stage. The mixture of this fluid, during its rise to the surface, with marine fluids, causes its cooling and evolution into two aqueous fluids; liquid (L2) and vapor (V) at Th = 300–450 °C, and $X_{NaCl}$ = 10–20 Wt % eq. NaCl. This mixture represents the beginning of the deposition of the primary sulfide paragenesis with a quartz gangue, (ii) a hypersaline fluid of a low temperature (brine Sf: Th = 120 to 160 °C and $X_{NaCl}$ = 30 to 35 Wt% eq. NaCl), which would be associated with the deposition of fluorite and (iii) a late aqueous fluid ($L_{Ca}$: $X_{NaCl}$ = 10 to 20 Wt% eq. NaCl, Th = 200 to 250 °C) which would occurs at the deposition of late sulfide ore paragenesis, associated with carbonate gangue, founded as traces in quartz and fluorite (Figure 24);

vi.   The early mineralizing event is linked to the placement of mafic rocks, during the development of continental rifting due to the Devono–Dinantian extension, while those at the origin of fluorite and late sulfides are associated either with the

tardi-Hercynian felsic magmatism of El Hammam, or with the effect of the thermal flux of anorogenic Triaso-Jurassic volcanism.

**Author Contributions:** Conceptualization, H.M. and M.A.; data curation, H.M. and A.M.; formal analysis, H.M., M.A., A.M., A.E.B. and A.T. and E.B.; funding acquisition, H.M., S.M. and M.E.A.; investigation, H.M., M.A. and A.M.; methodology, H.M., M.A., A.M., S.M., A.T., A.E.B. and E.B.; project administration, H.M. and M.A.; resources, H.M. and M.A.; software, H.M., M.E., A.E.B., A.M., A.-a.K. and S.M.; supervision, H.M. and M.A.; validation, H.M. and M.A.; visualization, H.M.; writing—original draft, H.M. and M.A.; writing—review and editing, H.M., M.A., M.E., S.M., A.T, A.-a.K. and E.B. All authors have read and agreed to the published version of the manuscript.

**Funding:** This research received no external funding.

**Data Availability Statement:** Data supporting the findings of this study are available upon request from the corresponding author, (Hafid Mezougane).

**Conflicts of Interest:** The authors declare no conflict of interest.

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
