# Peer review of "Sulfide and Fluoride Mineralization of the NNE Region of Achemmach (Central Morocco): Paragenetic Sequences and Pyrrhotite-Sphalerite Geothermometry Constraints"

_minerals, doi:10.3390/min12070790_

Round 1
Reviewer 1 Report
This paper reports detailed analysis results of sulfide and fluoride mineralization at NNE Achemmach, including mineral paragenesis based on optical microscopy and field observation, fluid inclusion data and EMPA analysis of individual minerals. These analyses are systematic, and the reports are detailed and well-organized. However, my main criticisms are : (1) I found the English in this manuscript is somehow poor and needs to be improved for better clarity. Typos are here and there. Especially some paragraphs and sentence in ‘Discussion’ need a major overhaul as the grammar is poor and it is often hardly to understand what the authors want to express. (2) References are missing for many statements in the ‘Discussion’ section. (3) The authors suggested that demixing of a higher T-P magmatic leads to L and V fluids of lower T-P and salinity. Yet, I doubt this claim as in theory, phase separation of a saline fluids of higher T-P due to depression should lead to a briny fluid and a dilute vapor phase (see Driesner’s papers for NaCl-H2O system). It is hard to imagine both L and V phases as claimed by authors to have lower salinity. Hence, a more careful explanation of such fluid evolution should be provided. A fluid mixing model of a high T-P saline magmatic fluid with lower T-P dilute meteoric water seems more likely to explain the observed data of this study. (4) Sulfide chemistry accounts for a significant portion of this paper but the interpretation is yet poor and the data is not fully discussed in the ‘Discussion’. Fluid evolution based on fluid inclusion data is currently the focus of the ‘Discussion’. So, I suggest author to pay more attention to the sulfide chemistry here as well if they deem the data relevant to the story in a better way, rather than just simply ‘show’ the data.
Overall, I suggest the paper should be of interests to a wide range of researchers in the filed as it clearly documents the ore genesis, fluids evolution and tectonic setting of a case deposit that show oceanic origins. I recommend the paper to be published after a moderate revision to improve the overall clarity and coherence of the observed geological evidence.
Author Response
Responses to reviewer’s suggestions/comments
We thank the Reviewer#1 for his pertinent remarks and suggestions which allowed us to improve significantly our manuscript and which opened a field of reflections for the future research work. The answer to each comment is given below (in red).
This work focused on Sulphide and fluoride mineralization of the NNE region of Achemmach (central Morocco): paragenetic sequences and pyrrhotite-sphalerite geothermometry constraints.
The first main goal of this work is to elucidate the spatio-temporal and genetic relationships that could exist between the sulphide-fluoride mineralizations and the magmatic host rocks (dolerites, gabbros, and pillow-lavas) recognized for the first time in the NNE Achemmach sector (Mezougane et al 2022) and their possible relationships with the hypothetical Achemmach granite.
The second objective is temperatures of the paragenetic stages are estimated on the basis of the geothermometry constraints of the mineralogical assemblages, in particular the pyrrhotite-sphalerite equilibrium, in which the FeO content varies from 9.23 to 14.42 Wt%, and the full study of their corresponding fluid phases. The following issues need to be addressed:
Point 1: I found the English in this manuscript is somehow poor and needs to be improved for better clarity. Typos are here and there. Especially some paragraphs and sentence in ‘Discussion’ need a major overhaul as the grammar is poor and it is often hardly to understand what the authors want to express.
Response 1: English improved.
Point 2: References are missing for many statements in the ‘Discussion’ section.
Response 2: References completed.
Point 3: The authors suggested that demixing of a higher T-P magmatic leads to L and V fluids of lower T-P and salinity. Yet, I doubt this claim as in theory, phase separation of a saline fluids of higher T-P due to depression should lead to a briny fluid and a dilute vapor phase (see Driesner’s papers for NaCl-H2O system). It is hard to imagine both L and V phases as claimed by authors to have lower salinity. Hence, a more careful explanation of such fluid evolution should be provided. A fluid mixing model of a high T-P saline magmatic fluid with lower T-P dilute meteoric water seems more likely to explain the observed data of this study.
Reponse 3: we agree with a fluid mixing model of a high T-P saline magmatic fluid with a lower T-P dilute meteoric fluid, as suggested by the reviewer (see modifications in text).
Point 4: Sulfide chemistry accounts for a significant portion of this paper but the interpretation is yet poor and the data is not fully discussed in the ‘Discussion’. Fluid evolution based on fluid inclusion data is currently the focus of the ‘Discussion’. So, I suggest author to pay more attention to the sulfide chemistry here as well if they deem the data relevant to the story in a better way, rather than just simply ‘show’ the data.
Response 4: Sulphide chemistry compositions was used on the one hand to support the papragenetic study and on the other hand to estimate, as a complement to the fluid inclusion study, the thermobarometric conditions.

Reviewer 2 Report
The manuscript needs a careful editing for occasional minor errors, e.g., page 3/33, line 92 and elsewhere.
Author Response
Responses to reviewer’s suggestions/comments
We thank the Reviewer#2 for his pertinent remarks and suggestions which allowed us to improve significantly our manuscript and which opened a field of reflections for the future research work. This work focused on Sulphide and fluoride mineralization of the NNE region of Achemmach (central Morocco): paragenetic sequences and pyrrhotite-sphalerite geothermometry constraints.
The first main goal of this work is to elucidate the spatio-temporal and genetic relationships that could exist between the sulphide-fluoride mineralizations and the magmatic host rocks (dolerites, gabbros, and pillow-lavas) recognized for the first time in the NNE Achemmach sector (Mezougane et al 2022) and their possible relationships with the hypothetical Achemmach granite.
The second objective is temperatures of the paragenetic stages are estimated on the basis of the geothermometry constraints of the mineralogical assemblages, in particular the pyrrhotite-sphalerite equilibrium, in which the FeO content varies from 9.23 to 14.42 Wt%, and the full study of their corresponding fluid phases.

Reviewer 3 Report
Sulphide and fluoride mineralization of the NNE region of Achemmach (central Morocco): paragenetic sequences and pyrrhotite-sphalerite geothermometry constraints
Hafid Mezougane, Mohamed Aissa, Souiri Muhammad, Azizi Moussaid, Abdelaziz El Basbas, Ahmed Touil, Essaid Bilal and Kharis Abdel-ali
In this article the authors present the results of the study of sulphide and fluorite mineralization that occur in an area located NE from Central Hercynian Morocco to NNE Achemmach. The study of this area, given the existence of some similar characteristics, can contribute to the knowledge of the origin of the fluorite and sulphide mineralizations in Central Morocco, as it occurs to the E of the important and REE-Rich fluorite deposit of El Hammam. The mineralizations studied are located either in Visean sedimentary formations or in magmatic bodies. The latter are described for the first time and correspond to pillow lava, dolerites and olivine gabbros.
The mineralization is multiphase and would be the result of the succession of three events: (i) a first high temperature hydrothermal sulphide event and a quartz gangue. (ii) The second event corresponding to a lower temperature fluorite hydrothermal event is identical to the El Hammam fluoride event (iii) a third one marked by the deposition of late sulphides associated with a carbonated gangue
The authors estimate the temperature conditions by studying fluid inclusions and studying the equilibrium between sulphides of the first event.
Although the mineralizations contain fluorite and IF studies were carried out in them, which are compared with those carried out by other authors for El Hammam, no information is given about this mineral, particularly with regard to its REE contents.
The studied area is formed by two types of Paleozoic rocks: (i) a metasedimentary sequence, composed of Middle Visean limestone and Upper Visean-Namurian schist-sandstone and flyschoid facies, and (ii) a series of magmatic rocks represented by volcanic rocks (lava pillow), hypovolcanic rocks (dolerites) and olivine- gabbros.
Pillow lavas occur in thin layers with pelitic sedimentary intercalations, indicating recurrent volcanic activity in short episodes at the end of the Middle to Upper Visean and before the Upper Visean- Namurian flysch (Fig 4 and Fig 5)
The dolerites are deformed, metamorphosed and altered and occur in dykes that cut through the Middle to Upper Visean shale-sandstone formations.
Th Gabbros with olivine, outcropping in varying dimensions (few to 20 m) with all minerals exhibiting different degrees of replacement by secondary minerals.
However, despite the importance given by the authors to the occurrence of these rocks, whose occurrence is described for the first time, no geochemical data are provided that allow their classification, which is important for a definition of a geodynamic context.
The studied area as the El Hammam area occurs in the Central Massif of the Moroccan Meseta which corresponds to a SW segment of the Variscan belt of Europe exhibiting a generalized stratigraphic column comprising Visean-Namurian sedimentary and volcaniclastic rocks. These rocks are folded and metamorphosed and in El Hammam area locally intruded by late Hercynian granitic stocks and dyke swarm of microgranites and tholeiitic dolerites attributed to the Upper Visean-Namurian (Jebrak, 1984).
The abundance of sphene and rutile in close association with hydrothermal alteration is interpreted by the authors because of low-temperature hydrothermal metamorphism of the seafloor. The argument is based only on petrographic observations and there are no geochemical elements to prove it. The existence of saline fluids in the mineralization may be related to an in-put of seawater and hence the controversy that still exists regarding the origin of fluorite mineralization.
From a fluid inclusion study similar to the one carried out by the authors, relatively low formation depths (<2km) were estimated for the fluorites of El Hammam (Bouabdellah et al 2016).
What is the estimated depth for the formation of mineralizations in the studied area? You calculated some isochores?
Geochronological data are very scarce. A. Cheilletz et al. 2010 proposed a 40Ar/39Ar age of 205±1 Ma and this age is significantly important as it improves the understanding of the genesis of El Hammam and undermined the hypothesis that fluorite mineralizations were related to Variscan granitic magmatism. The authors suggest that the 205±1 Ma age relates the El Hammam fluorite mineralization to Triassic–Jurassic rifting.
In Morocco numerous F-Ba-Base metal vein deposits are mainly of post-Carboniferous to Early Cretaceous in age (Bouabdellah et al. 2014).
The origin of REE -rich fluorite deposits remain controversial. Triassic-Jurassic anorogenic tholeiitic magmatism as proposed by Cheilletz et al (2010) is considered unsuitable for generating fluorite mineralization by Bouabdellah et al (2016) who considered El Hammam to be related to Pangean rifting and subsequent opening of the Central Atlantic during the period Permian-Triassic suggesting that there was little involvement of the tholeiitic magmatism other than to provide increased heat and fluid flow.
A Late Triassic-Early Jurassic volcanic sequence of the Central Atlantic Magmatic Province (CAMP) although corresponding to a Large Igneous Province (LIP) formed in continental environment, it contains subaqueous lava flows, including dominant pillowed flows but also occasional sheet flows. Intracontinental rifting, leading to Pangea break-up and Central Atlantic opening, initiated during latest Permian-earliest Triassic times and propagated northward along the trend of the Variscan orogen. Pangea rifting was accompanied by CAMP extrusion a LIP mostly composed of low titanium tholeitic basalts. CAMP magmatism is represented by the remains of intrusive rocks (crustal underplating, layered intrusions, sills and dykes) and less abundant extrusive rocks, mostly lava flows but also pyroclastic deposits (El Ghilani et al. 2017 and references)
In discussing the results, the authors write "Before discussing the petrological and geodynamic significance of igneous rocks, it is necessary to redraw some of their evidence: (i) the three recognized magmatic facies are the main constituents of an oceanic crust (gabbros at the base, doleritic dykes in the intermediate unit (blade dikes) and basaltic lava flows at the top. The magmatic facies are those of metamorphism (greenschist facies) and hydrothermal alteration of the ocean floor (serpentinization, spilitization, chloritization, epidotization, etc.); (iv) the geochemical affinity of these rocks is that of an alkaline magmatism with a tendency to differentiation".
But pillow lavas occur in thin layers with pelitic sedimentary intercalations, indicating recurrent volcanic activity in short episodes at the end of the Middle to Upper Visean and before the Visean-Upper Namurian flysch (see figure ).
The dolerites are deformed, metamorphosed and altered and occur in dykes that cut through the Middle to Upper Visean shale-sandstone formations and the olivine Gabbros, outcrop with all minerals exhibiting different degrees of replacement by secondary minerals.
"The early mineralizing event would be linked to the placement of mafic rocks, during the development of continental rifting due to the Devono-Dinantian extension, while those at the origin of fluorite and late sulphides would be associated either with El Hammam's Hercynian felsic magmatism, or with the effect of the thermal flow of Triaso-Jurassic anorogenic volcanism"
These conclusions are very confusing. What facts are they based on?
Is the age of the Oceanic Crust in your model Devono-Dinantian?
What age do the authors assume for the sulphide and fluorine mineralization of the NINE Achemmach region?
In our opinion a lot of work is still needed for this article to be publishable. Authors should try to answer the main questions that are posed both in these comments and throughout the text of the attached pdf.
References
Jébrak, M. (1985) Contribution à l’histoire Naturelle des Filons (F, Ba) du Domaine Varisque. Essai de Caractérisation Structurale et Géochimique Des Filons En Extension et En Décrochement Dans Les Massifs Centraux Français et Marocains; 473 p;
Mohammed Bouabdellah & David Banks & Andreas Klügel. (2010) Comments on “A late Triassic 40Ar/39Ar age for the El Hammam high-REE fluorite deposit (Morocco): mineralization related to the Central Atlantic Magmatic Province?” by Cheilletz et al. (Mineralium Deposita 45:323–329, 2010). Miner Deposita (2010) 45:729–731
DOI 10.1007/s00126-010-0288-5
Mohammed Bouabdellah; Oussama Zemri; Michel Jébrak; Andreas Klügel; Gilles Levresse; L. Maacha; Abdelaziz Gaouzi; Mohamed Souiah (2016)
Geology and Mineralogy of the El Hammam REE-Rich Fluorite Deposit (Central Morocco): A Product of Transtensional Pangean Rifting and Central Atlantic Opening. In book: Mineral Deposits of North Africa
DOI: 10.1007/978-3-319-31733-5_12
El Ghilani, S., Youbi N., Madeira J., Chellai E.H., Lopez-Galindo A., Martins L., Mata J. (2017) Environmental implication of subaqueous lava flows from a continental Large Igneous Province: Examples from the Moroccan Central Atlantic Magmatic Province (CAMP. Journal of African Earth Sciences 127 (2017) 211-221

Author Response
Responses to reviewer’s suggestions/comments
We thank the Reviewer#3 for his pertinent remarks and suggestions which allowed us to improve significantly our manuscript and which opened a field of reflections for the future research work. The answer to each comment is given below (in red).
This work focused on Sulphide and fluoride mineralization of the NNE region of Achemmach (central Morocco): paragenetic sequences and pyrrhotite-sphalerite geothermometry constraints.
The first main goal of this work is to elucidate the spatio-temporal and genetic relationships that could exist between the sulphide-fluoride mineralizations and the magmatic host rocks (dolerites, gabbros, and pillow-lavas) recognized for the first time in the NNE Achemmach sector (Mezougane et al 2022) and their possible relationships with the hypothetical Achemmach granite.
The second objective is temperatures of the paragenetic stages are estimated on the basis of the geothermometry constraints of the mineralogical assemblages, in particular the pyrrhotite-sphalerite equilibrium, in which the FeO content varies from 9.23 to 14.42 Wt%, and the full study of their corresponding fluid phases. The following issues need to be addressed:
Point 1: "This magmatic activity generated, directly or indirectly, the deposition of several types of mineralization of economic interest".What do you mean by "directly or indirectly"?
Response 1: corrected (see the manuscript).
Point 2: The authors make an exhaustive reference to different deposits in the central massif of Morocco but do not present a map with the location of these deposits (In Fig.1 only the location of three deposits is indicated). References are also not presented, making it impossible for a reader unfamiliar with the geology and metallogeny of Morocco to follow what they intend to convey.
Response 2: corrected (see the manuscript).
Point 3: The authors locate the area to be studied in the NNE of the Achemmach tin deposits. However, they do not specifically mention Sn mineralizations and the "hypothetical granite they relate to". Are there works that refer to the connection of Achemamach tin mineralizations with a granite? (Reference?)
Response 3: corrected (see the manuscript).
Point 4: the granites are calc-alkaline (not calcareous) and mostly peraluminous.
Response 4: corrected (see the manuscript).
Point 5: Three cross-sections are shown on the map, but then they do not show any profile.
Response 5: corrected (see the manuscript).
Point 6: Microscopic ? (Figures 7 and 9)
Response 6: these are the macroscopic samples ( a and b for figure 7, a for figure 9)
Point 7: "The abundance of titanic minerals such as sphene and rutile in close association with the aforementioned hydrothermal alteration confirms the presence of an evident seafloor hydrothermal metamorphism in this area."
Response 7: references : (Coogan et al.2001, Gillis et al.2003, Christian Nicollet, Fanny Cattani 2018, C.Dupuis et al.2005)
Point 8: The Fe-Cu-Zn-Pb sulfide mineralization of the NNE Achemmach sector is housed both in Visean and Upper-Middle Age Visean and High-Namurian sedimentary formations, as well as in magmatic formations. It is essentially present in the disseminated form, but also in the form of veins and veins. (Figure 12). (There is a repetition).
Response 8: corrected (see the manuscript)
Point 10: In this table are indicated the (.): average values; [−]: and the minimum and maximum values. However, the indicated average values are greater than the maximum values ??
Response 10: Corrected (see table 2)
Point 11: The final melting temperatures in Lf -5 to -20ºC and in LCa -10 to -15 are different. How you justify the similarity in %NaCl?
Response 11: Tmi -5 to -20°C in Lf corresponds to the whole range of variation of these temperatures, while the mode, identical to the Tmi of Lca, is between -10 and -15°C.
Point 12: These two fluids have very different characteristics. Sf Th 120-160 ºC and salinity 30-35%NaCl and LCa Th 200-250ºC an salinity 10 to 20%NaCl. With which of the two is the deposition of sulfides associated?
Response 12: Lf (150 to 250°C) is the fluid which has properties to the fluid LCa (200 to 250) and not Sf. (Corrected in the text).
Point 12: Did you analyze the studied rocks? You should put some references since geochemical data are not shown.
Response 13: corrected (see the manuscript)
Point 13: The discussion that they present and that leads you to propose that you are facing a rifting situation is surely based on references that should support your model. References must be provided.
Response 13: corrected (see the manuscript)
Point 14: Don't they give an explanation for why the fluids they assume to be later are of higher temperature than those that presided over the origin of fluorite?.
Response 14: at Achemmach the emplacement of felsic magmatism is polyphase (Aissa 1997). The hydrothermal event at the origin of the carbonatation and the late sulphides would be linked to a late magmatic event hotter and different from the one at the origin of the fluorite, slightly lower temperature.
Point 15: All these are arguments to associate this early hydrothermal event with the mafic magmatism of this sector, a second low temperature hydrothermal event associated with fluorite, and a third later event that would be the cause of the late deposition of sulfide paragenesis and the associated carbonation. These last two hydrothermal events are similar to those linked: a) either to felsic magmatism as is classically admitted by many authors for fluorite and sulfides from El Hammam (???); b) or to the effect of the heat flux generated by the basic volcanism in connection with the Triassic-Jurassic opening of the Atlantic Ocean proposed by [59] after the dating of the fluorite mineralization of El Hammam by the 40Ar/39Ar method in potassium feldspar.
Point 16: I don't understand the relation between two events separate by several MA.
Response 16: there is no temporal relationship between these two events, just a spatial relationship . They are 2 surimposed events, but separeted in time and recorded by the same host minerals.

Round 2
Reviewer 3 Report
Dear Authors
After reviewing the manuscript, I think that you considered my suggestions and made several corrections to the initial manuscript. Congratulations on your effort
In the present version there are several small typos that should be corrected (p.e. line 131 “The Lower Permian (Autunian) formationsare formedby conglomerates”.
There are also, for example in line 80, two references (Kosakevitch 1972 and Charty, 1980) that must be indicated by the respective numbers [3] and [4].
You should carefully read the entire latest version and correct these minor errors.
Good luck